# DYNAMIC ELIMINATION FOR PAC OPTIMAL ITEM SELECTION FROM RELATIVE FEEDBACK

## ABSTRACT

We study the problem of best-item identification from relative feedback where a learner adaptively plays subsets of items and receives stochastic feedback in the form of the best item in the set. We propose an algorithm - *Dynamic Elimination* (DE) - that dynamically prunes sub-optimal items from contention to efficiently identify the best item and show a strong sample complexity upper bound for it. We further formalize the notion of *inferred updates* to obtain estimates on item win rates without directly playing them by leveraging item correlation information. We propose the *Dynamic Elimination by Correlation* (*DEBC*) algorithm as an extension to *DE* with inferred updates. We show through extensive experiments that *DE* and *DEBC* vastly outperforms all existing baselines across multiple datasets in various settings.

## 1 INTRODUCTION

Learning to rank from feedback about a set of items is an important problem in machine learning with applications in many areas including sociology (Vieira et al., 2007; Zareie & Sheikhahmadi, 2018), information retrieval (Hofmann et al., 2013; Grotov & De Rijke, 2016; Guo et al., 2020), search engine optimization (Kakkar et al., 2015; Krrabaj et al., 2017), recommender systems (Balakrishnan & Chopra, 2012; Tang & Wang, 2018; Bałchanowski & Boryczka, 2023), and, more recently, natural language generation (Hofstätter et al., 2023; Zhang et al., 2023; Chuang et al., 2023). An important sub-problem is learning to rank from relative feedback (Chen et al., 2018; Saha & Gopalan, 2019c; Haddenhorst et al., 2021). In this setting, a set of items are played and stochastic relative feedback is received in the form of the best item or a full or partial ranking of the items.

We consider the problem where we play fixed-sized item subsets and receive relative feedback modelled by the Plackett-Luce (PL) model with the aim of PAC-learning the best item. Existing works in this setting (Saha & Gopalan, 2019a;b) including instance-optimal algorithms (Saha & Gopalan, 2020b; Haddenhorst et al., 2021) typically evaluate a static item subset and retain only the set winner before moving on to the next. However, subset plays are wasted on items in the subset that are already known to be suboptimal before the set winner is determined. We investigate if flexible item elimination is feasible to alleviate this inefficiency.

Furthermore, no assumption is usually made about the underlying feedback distribution beyond some random utility model. However, we argue that information about the entities to be ranked (e.g. items in recommender systems, documents in the information retrieval setting, nodes in a social network, etc.) is often readily available. Motivated by this, we investigate the question: *Given what we know about items $i$, $j$ and $k$, if item $i$ is ranked above/below item $k$, how likely is it that item $j$ is ranked above/below item $k$?*

Latent embedding models are commonly used in many domains, including natural language processing (Pennington et al., 2014; Church, 2017), information retrieval (Zuccon et al., 2015; Palangi et al., 2016) and recommender systems (Chen et al., 2019; Huang et al., 2020), to flexibly represent unstructured information as vectors in a latent space such that the vectors of closely related items are highly similar. We apply the latent embedding model to the PL model such that item latent scores are given by query-item vector cosine similarity and aim to learn a PAC-best item from stochastic relative feedback. Our contributions are fourfold:

1. We propose an algorithm *Dynamic Elimination* (*DE*) for the $(\epsilon, \delta)$ PAC best-item objective with sample complexity $O(\frac{n}{\epsilon^2} \ln(\frac{n}{n_s \delta}))$ based on flexibly eliminating items once they are deemed suboptimal. *DE does not* leverage correlation information.

2. We formalize the notion of *inferred updates* - probabilistic updates to the estimates of item pairwise win ratios by observing the win rates of related items - and prove that the sample mean of an inferred update sequence constitutes an unbiased estimator.

3. We propose the *Dynamic Elimination by Correlation* (*DEBC*) algorithm as an extension to *DE* that leverages item information in the form of an item vector correlation matrix. We show a sample complexity of $O\left(\max\left(\frac{R}{\epsilon^2}\ln(\frac{n}{n_s\delta}), \frac{n^*}{\epsilon^2}\ln(\frac{n^*}{n_s\delta})\right)\right)$ with a noisy $R$-Block-Rank item correlation structure.

4. We demonstrate through experiments across multiple datasets in various settings that both *DE* and *DEBC* outperform all existing SOTA benchmarks by over an order of magnitude in sample complexity without loss of accuracy.

## 2 RELATED WORK

Reward maximization from sampling an unknown reward distribution has been extensively studied in the classical multi-armed bandit setting where an absolute stochastic reward is observed (Even-Dar et al., 2006; Scott, 2010; Agrawal & Goyal, 2012). This was extended to relative feedback in the duelling bandit problem (Yue et al., 2012) which has been the object of a large body of work (Dudík et al., 2015; Chen & Frazier, 2017; Jamieson et al., 2015), including extensions to multiwise comparisons (Brost et al., 2016; Sui et al., 2017; Saha & Gopalan, 2019b). Beyond regret minimization in the bandit setting, active arm ranking or learning of the best arm has been studied both in the exact (Jamieson & Nowak, 2011; Maystre & Grossglauser, 2017; Ren et al., 2019; 2021) and PAC setting (Saha & Gopalan, 2019a; Agarwal et al., 2022). In particular, Saha & Gopalan (2019c) and Saha & Gopalan (2019a) present algorithms for obtaining the PAC best item and full ranking respectively under a PL model assumption with fixed sized subsets which is identical to our setting. (Saha & Gopalan, 2020b) and (Haddenhorst et al., 2021) propose instance optimal algorithms which outperform the former in empirical trials. More recently, Yang & Feng (2023) proposed an algorithm in a setting where subsets of variable size can be played.

However, these algorithms often require up to millions of samples to rank only a few items. The inefficiency lies in statically evaluating a subset to determine the winner before moving on to a new subset. This means that a set containing two closely matched items can be "stuck" for many turns, wasting item subset plays on the other clearly suboptimal items in the subset. We propose dynamic item elimination to solve this problem.

Furthermore, ranking algorithms typically do not leverage additional information about the underlying reward distribution to improve performance. The body of work in this area is surprisingly relatively small. Sui et al. (2017); Saha & Ghoshal (2022) consider arms with correlated rewards while (Gopalan et al., 2016) considers a contextual bandit setting where user preferences are latent mixtures of a set of reward distributions. While learning to rank items by assuming latent vector representations has been widely studied across many domains (Balakrishnan & Chopra, 2012; Palangi et al., 2016; Zuccon et al., 2015), it is very limited in this setting. To this end, (Chen & Frazier, 2016; Mesaoudi-Paul et al., 2020) assume random utility models where the latent scores are derived from the item vectors and an unknown context vector. Jamieson & Nowak (2011); Chen & Frazier (2016) suggest algorithms for precise ranking based on pairwise feedback assuming a latent reward given by query vector-item vector Euclidean distance. However, the algorithms are heavily reliant on complete knowledge of the exact vector representations, which can be limiting in real-world scenarios. In comparison, we utilize cosine similarity as a vector distance which is widely used across all machine learning domains and only require the item correlation matrix as an input instead of the exact item vectors.

## 3 PRELIMINARIES AND PROBLEM SETUP

**Notation** Before proceeding, we establish some notation. We use $[n]$ to denote the set $1, 2, ..., n$. $|S|$ denotes the cardinality of a set $S$. We use $Pr(A)$ to denote the probability of event A in a probability space that will be clear from context. In particular, $Pr_{\mathbf{q}}(...)$ denotes the probability space over all

possible vectors $\mathbf{q}$. We denote the probability that an item $i$ beats an item $j$ as $p_{ij} = Pr(i|\{i,j\})$. pdf(X) denotes the probability distribution of some random variable $X$ and pdf(X|Y) denotes the conditional distribution of $X$ given $Y$. $\mathbf{1}(\varphi)$ denotes an indicator variable that assumes the value 1 if the predicate $\varphi$ is true and 0 otherwise.

**Feedback Model**   We consider the best-item identification problem from subset wise relative feedback drawn from a reward distribution modelled on a PL model. Formally, we consider a set of $n$ items $[n] := \{1, 2, ..., n\}$; each turn, the learner plays a set of $n_s$ items $S_t \subseteq [n]$ and receives $i_t \in S_t$ as the best item with probability given by $Pr(i_t = i|S_t) = \frac{\theta_i}{\sum_{j \in S_t} \theta_j}$ where $\theta_i$ is the latent score for item $i$. A choice model is said to fulfil Independence of Irrelevant Attributes (IIA) if for any two sets $S_1, S_2 \ni i_1, i_2$ containing items $i_1, i_2 \in [n]$, $\frac{Pr(i_1|S_1)}{Pr(i_2|S_1)} = \frac{Pr(i_1|S_2)}{Pr(i_2|S_2)}$, i.e. the ratio of the winning probabilities of the two items is independent of other items in the set (Benson et al., 2016). The defined PL model clearly fulfils this criteria.

**Performance Objective: $(\epsilon, \delta)$-PAC best-item**   Clearly, such a formulation admits the existence of a Condorcet winner which is the item with the highest latent score, i.e. $i^* = \text{argmax}_{i \in [n]}(\theta_i)$. By the IIA property, we have that $p_{i^* i} > \frac{1}{2}$ $\forall i \in [n] \setminus \{i^*\}$. WLOG, we denote this item by $1 = i^*$. An item is said to be $\epsilon$-optimal if the probability that it beats the winning item 1 is larger than $1/2 - \epsilon$, i.e. $Pr(i|\{i, 1\}) > 1/2 - \epsilon$. A sequential algorithm is said to be $(\epsilon, \delta)$-PAC (probably approximately correct) if within a finite number of subset plays it stops an outputs an item with probability 1 and if the item is $\epsilon$-optimal with probability at least $1 - \delta$. The number of subset plays before stopping is the algorithm sample complexity.

## 4   ESTIMATING PAIRWISE WIN RATIOS FROM RELATIVE FEEDBACK

A common approach to item-ranking with relative feedback is to employ rank breaking and maintain a preference matrix that tracks the empirical win ratios, i.e. the rate at which an item is selected over the other. In rank breaking, partial rankings are decomposed into pairwise comparisons and pairwise win ratios are estimated independently (Saha & Gopalan, 2019c). The IIA property of the PL model allows the use of rank breaking. We use the term *empirical updates* to refer to preference matrix updates arising directly from user feedback as opposed to *inferred updates* which will be covered in Section 6.

Formally, let us denote the preference matrix at iteration t by $\mathbf{P}(t) \in \mathbb{R}^{n \times n}$, and the number of times an item $i$ has won a set containing $S$ as a subset as $n_{i|S}(t)$. Then, we have $P_{ij}(t) = \frac{n_{i|\{i,j\}}(t)}{n_{i|\{i,j\}}(t) + n_{j|\{i,j\}}(t)}$. Given a sequence of sets that have been played by the learner up to timestep $t$ $\mathcal{S}(t) = \{G(\tau) : \tau = 1, 2, ..., t\}$ and a sequence of winning items $\iota(t) = \{i_\tau : \tau = 1, 2, ..., t\}$, let us consider for some item pair $i, j$ the subsequence of winners $\iota_{ij}(t) = \{\mathbf{1}(i_\tau = i) : \tau \in [1, t], i_\tau \in \{i, j\}\}$ for which the winner is either $i$ or $j$. As shown in (Saha & Gopalan, 2019c;b;a; Saha & Gaillard, 2022), we can treat this binary subsequence as a sequence of iid Bernoulli random variables with success parameter $p_{ij}$ due to the IIA property. Consequently, $P_{ij}(t)$ is an unbiased estimator for $p_{ij}$ with bounded deviation according to Hoeffding's Inequality.

## 5   ALGORITHM: DYNAMIC ELIMINATION

### 5.1   ALGORITHM OVERVIEW

We propose the *Dynamic Elimination* (*DE*) algorithm as a direct replacement for existing PAC-best item algorithms under a PL model assumption (Saha & Gopalan, 2019c; 2020b; Haddenhorst et al., 2021). It progressively removes items from contention once they are no longer potential Condorcet winners.

During each iteration, an item subset is played (initialized randomly in Alg. 1: 2-4) and the preference matrix is updated via rank breaking (Alg. 1: 6-8). The item subset is then updated as follows: When items are deemed suboptimal with high probability, they are removed (Alg. 2: 1-7). An item that is not eliminated after a certain number of plays becomes a potential replacement to the running winner. It replaces the running winner if it is the highest probability replacement, inheriting the wins/losses

---

**Algorithm 1:** Dynamic Elimination (*DE*)

---

**Input:** set of items: $[n]$, subset size: $n_s$, error bias: $\epsilon > 0$, confidence parameter: $\delta > 0$

**Initialize:** uneliminated item set: $S \leftarrow [n]$, item subset to play: $G \leftarrow \emptyset$, empirical pairwise win

ratio matrix: $\mathbf{W} \leftarrow [0]^{n \times n}$, $\gamma \leftarrow \left\lceil \frac{n}{n_s} \right\rceil$, $m \leftarrow \frac{2 \ln(\gamma/\delta)}{\epsilon^2}$

**1 while** $|S| > 1$ **do**

**2**    **if** $|G| < n_s$ **then**

**3**      $a \leftarrow$ random item from $S \backslash G$ // `randomly select unplayed item`

**4**      $G \leftarrow G \cup \{a\}$ // `build initial item subset/replenish eliminated item`

**5**    **if** $|G| = n_s$ **then**

**6**      Play set $G$, $i \leftarrow$ winning item

**7**      $\forall k \in G, k \neq i : W_{ik} \leftarrow W_{ik} + 1$ // `Update empirical pairwise win ratios`

**8**      $\mathbf{N} \leftarrow \mathbf{W} + \mathbf{W}^T, \quad \mathbf{P} = \mathbf{W}/\mathbf{N}$

**9**      $\mathbf{U} = \mathbf{P} + \sqrt{\frac{\ln(\gamma/\delta)}{2\mathbf{N}}}$ // `Update upper confidence bound matrix`

     // `run` *update-set* `to eliminate items, update running winner`

**10**      $G, S, i^* \leftarrow$ *update-set*$(G, i^*, \mathbf{U}, \mathbf{P}, \mathbf{N}, S, m, \epsilon)$

     // `keep only potential Condorcet winners`

**11**      $S \leftarrow \{j \in S : \min_{j' \in S} U_{jj'} \geq \frac{1}{2}\}$

**12**      $S \leftarrow S \backslash \{j \in S : P_{i^*j} \geq \frac{1}{2} - \frac{\epsilon}{2} \text{ and } N_{i^*j} \geq m\}$

---

of the outgoing winner (Alg. 2: 8-11); otherwise, it is eliminated. Removed items are replaced by randomly selected items (Alg. 1: 3, Alg. 2: 5).

The main innovations are listed below. A discussion of their importance to the accommodation of inferred updates in *DEBC* can be found in Appendix D.1.

---

**Algorithm 2:** *DE update-set* subroutine - eliminates suboptimal items, updates item subset and running winner

---

**Input:** subset $G$, current winner $i^*$, upper confidence bound matrix $\mathbf{U}$, preference matrix $\mathbf{P}$, count matrix $\mathbf{N}$, potential candidate set: $S$, max no. of updates $m$, error bias $\epsilon$

**Initialize:** updated subset $H \leftarrow \emptyset$, potential running winner challengers

$W \leftarrow \{j \in G \backslash \{i^*\} : N_{i^*j} \geq m, P_{i^*j} < \frac{1}{2} - \frac{\epsilon}{2}\}$

**1 for** $j \in G \backslash (\{i^*\} \cup W)$ **do**

**2**    **if** $U_{ji^*} < 1/2$ **or** $N_{i^*j} \geq m$ **then**

     // `eliminate item if it is not a potential Condorcet winner`

**3**      $S \leftarrow S \backslash \{j\}$

**4**      $a \leftarrow$ random item from $S \backslash G$

**5**      $H \leftarrow H \cup \{a\}$ // `replace with randomly selected item`

**6**    **else**

**7**      $H \leftarrow H \cup \{j\}$

// `update current running winner` $i^*$ `with new running winner` $i$

**8 if** $|W| \neq 0$ **then**

**9**    $i \leftarrow \arg\max_{j \in W} P_{i^*j}$ // `item with highest win prob. over current winner` $i^*$

   // `the incoming running winner inherits the win/losses from the`

     `outgoing winner as a conservative estimate`

**10**    $\forall j \in S \backslash \{i\} : P_{ij} \leftarrow P_{i^*j} \times N_{i^*j} + P_{ij} \times N_{ij}, \quad N_{ij} \leftarrow N_{ij} + N_{i^*j} \; i^* \leftarrow i$

**11**    $H \leftarrow H \cup W$

**12 else**

**13**    $H \leftarrow H \cup \{i^*\}, i \leftarrow i^*$

**Output:** $H, S, i$

---

**Dynamic item elimination**  Existing PAC algorithms typically play a set of items for a certain number of rounds before keeping the winning item and eliminating the rest (Ailon et al., 2012; Ailon, 2012; Saha & Gopalan, 2019c; 2020a; Haddenhorst et al., 2021). In contrast, *DE* eliminates an item once it is no longer a potential Condorcet winner (with high probability) and avoids the redundancy of playing an item that is known to be sub-optimal. We show that introducing this flexibility improves the worst case sample complexity (Theorem 1) and leads to vastly lower sample complexity in practice (Section 8).

**Running winner inheritance**  A challenge in accommodating flexible item elimination is that a running winner can potentially be eliminated before items that have received updates from it can be eliminated with certainty. This renders existing updates redundant since the items need to accumulate pairwise interactions with the new running winner. To avoid this, we allow the new running winner to inherit the pairwise interactions of previous running winners. We show in Lemma 10 that this constitutes a conservative estimate (i.e. the win ratio of the new running winner exceeds that implied by the inherited interactions with high probability).

## 5.2 Sample complexity and correctness of *DE* for the general case

As is the convention (Saha & Gopalan, 2019a; 2020a; Haddenhorst et al., 2021), we present sample complexity upper bounds for *DE*. We further present sample complexity lower bounds and an expected sample complexity under certain assumptions.

**Theorem 1 (Sample complexity and correctness of *DE* in the general case)** *DE is* $(\epsilon, \delta)$*-PAC with worst-case sample complexity* $O(\frac{n}{\epsilon^2} \ln(\frac{n}{n_s \delta}))$.

**Proof (sketch)**  To prove the correctness of *Dynamic item elimination*, we prove that the running winner $i_*$ is pairwise $\epsilon$-optimal with high probability to any items eliminated during its reign. We then prove the validity of *Running winner inheritance* by showing that the successor is optimal to the running winner it replaces with high probability. Combining both results allows us to prove the $\epsilon$-optimality of the winner completing the proof for correctness. We prove sample complexity by calculating the minimum item elimination frequency by considering all possible pairwise win count scenarios which then yields the maximum algorithm stopping time. The complete proof is given in Appendix E.4.

**Lemma 1 (Sample complexity lower bounds for DE)** *DE is* $(\epsilon, \delta)$*-PAC with best-case sample complexity* $O\left(\frac{n}{n_s} \ln\left(\frac{n}{n_s \delta}\right)\right)$.

**Remarks**  The best-case sample complexity corresponds to the case in which the eventual winner is selected in the initial item subset and continually wins all subset plays. The complete proof is in Appendix E.5.1.

**Lemma 2 (Expected sample complexity for DE)** *Given a reward distribution such that* $Var(p) = V$, *DE is* $(\epsilon, \delta)$*-PAC with an expected sample complexity upper bound of* $O\left(\frac{n(1-V)}{\epsilon^2} \ln\left(\frac{n}{n_s \delta}\right)\right)$.

**Remarks**  Since sample complexity is dependent on the latent reward distribution, we derive the expected sample complexity lower bounds as a function of the variance of the pairwise win probabilities $p_{ij}$ which we denote $Var(p)$. Intuitively, if $Var(p)$ is low, i.e. the pairwise win probabilities are generally close to $1/2$ and suboptimal items will not be easily eliminated. In this case, the expected sample complexity approaches the worst case sample complexity. The complete proof can be found in Appendix E.5.2.

## 6 Estimating pairwise win ratios with item correlations

In Section 4, we investigated how empirical updates can be employed to estimate pairwise win ratios. Here, we investigate how this can be extended to admit probabilistic updates to items that are not in the played set but sufficiently correlated to items in the set. We shall call these *inferred updates*.

## 6.1 LATENT EMBEDDING MODEL

We build upon the PL model described in Section 3 by assuming a latent item vector representation such that the latent scores are given by the cosine similarity between the item embeddings and an unknown query embedding. Formally, both the items and the query are represented by fixed $d$-dimensional latent vectors $\mathbf{v}_i \in \mathbb{R}^d$, and $\mathbf{q} \in \mathbb{R}^d$ respectively, and the latent scores are given by $\theta_i = e^{\mathbf{q} \cdot \mathbf{v}_i}$. We constrain both the query vectors and item vectors to have unit norm, i.e. $|\mathbf{q}| = 1, |\mathbf{v}_{i \in [n]}| = 1$. We assume that at least the item correlations are known to the user. We denote the item correlation matrix by $\mathbf{C} \in \mathbb{R}^{n \times n}$ where $C_{ij} = \mathbf{v}_i \cdot \mathbf{v}_j$.

## 6.2 CONDITIONAL PROBABILITIES OF CORRELATED ITEM LATENT SCORES

To extend empirical updates to inferred updates on items outside the played set, let us define the win ratio conditional probability $p_{jk|ik}$ as $p_{jk|ik} = Pr_{\mathbf{q}}\left(p_{jk} > \frac{1}{2} \mid p_{ik} > \frac{1}{2}\right)$.

**Theorem 2 (Conditional probabilities of win ratios)** *Given items* $i, j, k \in [n]$*, the following holds true:*

$$p_{jk|ik} = p_{kj|ki} = 1 - \frac{1}{\pi}\cos^{-1}\left(\frac{\mathbf{v}_i \cdot \mathbf{v}_j - \mathbf{v}_i \cdot \mathbf{v}_k - \mathbf{v}_j \cdot \mathbf{v}_k + 1}{2\sqrt{(1 - \mathbf{v}_j \cdot \mathbf{v}_k)(1 - \mathbf{v}_i \cdot \mathbf{v}_k)}}\right) \tag{1}$$

**Proof (sketch)** The main intuition is to consider that all item/query vectors lie on a $d$-dimensional unit hypersphere and that a condition $p_{ij} > 1/2$ induces a partitioning of the hypersphere such that query vectors that fulfil this condition lie on a hyper-hemisphere. The joint probability is in turn given by the area of intersection between two hemispheres. Consequently, the conditional probability can be obtained using the chain rule. The full proof is given in Appendix E.1.

## 6.3 COMBINING INFERRED UPDATES WITH EMPIRICAL UPDATES

In this section, we discuss the incorporating of inferred updates as Bayesian updates. From Section 4, $P_{ij}(t)$ is an unbiased estimator for $p_{ij}$ by viewing the empirical observations as a sequence of iid. Bernoulli random variables. Since the Beta distribution is the conjugate prior to the Bernoulli distribution, following $|\iota_{ij}(t)|$ Bayesian update steps as follows:

$$\text{pdf}(p_{ij}|x_t \sim \text{Bernoulli}(p_{ij})) = \text{Beta}(\alpha + x_t, \beta + 1 - x_t), \quad p_{ij} \sim \text{Beta}(\alpha, \beta)$$

the posterior predictive distribution of $p_{ij}$ at timestep $t$ is given by

$$\text{pdf}(p_{ij}|\iota_{ij}(t)) = \text{Beta}(n_{i|\{i,j\}}(t) + 1, n_{j|\{i,j\}}(t) + 1)$$

To extend this to inferred updates, we interpret them as probabilistic observations, i.e. given a trial yielding an observation that item $i$ is preferred over item $k$, we consider that we have also observed that item $j$ is preferred over item $k$ with probability $p_{jk|ik}$. Then, an inferred update sequence for any item pair $j, k$ can be defined as

$$\iota_{ij}^*(t) = \prod_{i \in [n]} \mathcal{F}_{p_{ij|ik}} \iota_{ik}(t)$$

where the function $\mathcal{F}_p : \{0, 1\}^L \rightarrow \{p, 1 - p\}^L$ modulates a binary sequence by the probability $p$. $\prod$ denotes sequence concatenation. To incorporate this as a Bayesian update, we rely on Jeffrey's Conditionalization (Jeffrey, 1990; van Fraassen, 1986): $\text{pdf}(p_{jk} \sim \text{Beta}(\alpha, \beta) \mid Pr(x_t) = p_{jk|ik}) = p_{jk|ik} \times \text{Beta}(\alpha + 1, \beta) + (1 - p_{jk|ik}) \times \text{Beta}(\alpha, \beta + 1)$.

**Theorem 3 (Estimating $p_{ij}$ from inferred updates)** *For any item pair $i, j$, given a sequence of binary empirical updates $\iota_{ij}(t)$ and a sequence of inferred updates $\iota_{ij}^*(t)$, the sample mean*

$$P_{ij}(t) = \frac{1}{|\iota_{ij}(t)|}\sum_{x \in \iota_{ij}(t)} x + \frac{1}{|\iota_{ij}^*(t)|}\sum_{p \in \iota_{ij}^*(t)} p \tag{2}$$

*is an unbiased estimator of $p_{ij}$.*

**Proof (sketch)**   We jointly consider both empirical and inferred updates as a single sequence of probabilistic updates ($p = 0, 1$ for empirical updates) and show that this results in a Beta distribution mixture. We then prove that the mean of this distribution is in fact the sample mean. The full proof is in Appendix E.2.

**Combining inferred updates from multiple items**   While we note that jointly considering empirical and inferred updates breaks the identically distributed condition, we can can combine both into a single sequence by considering empirical and inferred updates as two separate stages and supplying the posterior distribution of the first stage as the prior distribution of the second stage. Consequently, inferred updates from multiple items forms a multi-stage update, with each item yielding a sequence of iid. updates forming a single stage. This is further discussed in Appendix B.1.

**Validity of considering inferred updates from multiple items separately**   It is essential to note that we consider the inferred updates from multiple items separately. While considering evidence from multiple item pairs jointly yields an optimal estimate, computing the higher-order probabilities is intractable. In Appendix B.2, we analyze the feasibility of considering only first-order conditional probabilities. We show that treating the inferred updates from multiple items independently and taking the mean of the first-order probabilities is a conservative estimate of the high order conditional probability when the constituent probabilities are high. Consequently, we employ the heuristic of weighting updates to assign higher importance to probabilities close to 1 (Appendix B.5).

## 7   ALGORITHM: DYNAMIC ELIMINATION BY CORRELATION

### 7.1   ALGORITHM OVERVIEW

We propose *Dynamic Elimination by Correlation* (*DEBC*) as an extension to *DE* that takes in an item vector correlation matrix as an input which it leverages for inferred updates (Section 6) to the preference matrix as well as item selection. The complete algorithm is in Appendix D.2.

**Item selection**   The main idea is to construct an initial set of items that are poorly correlated with each other to yield higher conditional probabilities (given items $i, j, k$, Eqn. 1 shows that $p_{jk|ik}, p_{kj,ki}$ increases for some fixed $\mathbf{v}_i \cdot \mathbf{v}_j$ as $\mathbf{v}_{i/j} \cdot \mathbf{v}_k$ decreases) and to maximize inferred updates by covering the largest possible item space. For the latter reason, we also select the item that is the most correlated with other items as the first running winner. This concept is extended to the replacement of eliminated items - items that are least correlated to items that have already been played are selected. This allows *DEBC* to sweep the largest item space in the fewest number of plays.

### 7.2   SAMPLE COMPLEXITY AND CORRECTNESS OF *DE* WITH *R-Block-Rank* ITEM CORRELATION

In the case where all inferred updates are insignificant, Theorem 1 also applies to *DEBC*. Instead, we consider a noisy $R$-Block-Rank instance similar to that in (Ghoshal & Saha, 2022). In the $(r, c, c')$ noisy $R$-Block-Rank model, the items can be partitioned into blocks $B_1 \uplus B_2 \uplus B_3 \ldots \uplus B_R$ such that the following holds: 1) Given any 2 items $i, j \in [n]$ from the same partition, i.e. $\exists r \in [1, R] : i, j \in B_r$, then the following must be true: $\mathbf{v}_i \cdot \mathbf{v}_j \geq c$. 2) Given any 2 items $i, j \in [n]$ that do not share a partition, i.e. $\nexists r \in [1, R] : i, j \in B_r$, then the following must be true: $\mathbf{v}_i \cdot \mathbf{v}_j \leq c'$.

**Validity of inferred updates**   We recall that the inferred updates are inherently probabilistic, dependent on conditional probabilities defined over the space of all query vectors. Importantly, for any inferred update based on $p_{jk|ik} \neq 1$, there will be a region of query vectors for which the inferred updates are consistently wrong and unlike empirical updates, this deviation will not be resolved by increased sampling. Consequently, the $(\epsilon, \delta)$-PAC condition cannot be met without imposing additional constraints.

**Theorem 4 (Sample complexity and correctness of *DEBC* with $R$-Block-Rank correlation)**
*Given that the item correlation follows a R-Block-Rank model and that the partition containing the winning item $B_1$ contains $n^*$ items, i.e. $|B_1| = n^*$, DEBC is $(\epsilon, \delta)$-PAC with worst-case sample*

*complexity*

$$O\left(\max\left(\frac{\max(R, n_s \ln(n_s))}{w_{\min}^{in}\epsilon^2}\ln(\frac{n}{n_s\delta}) \ , \ \frac{n^*}{\epsilon^2}\ln(\frac{n^*}{n_s\delta})\right)\right) \tag{3}$$

*given that the following conditions are met:*

1. $\mathbf{q} \cdot \mathbf{v}_1 \leq 1 - \varepsilon$

2. $(c - c')(1 - \varepsilon) - \sqrt{2\varepsilon - \varepsilon^2}\left(\sqrt{1 - c'^2} + \sqrt{1 - c^2}\right) > \ln\left(\frac{1+2\epsilon}{1-2\epsilon}\right)$

3. $1 - \frac{\delta n^*}{n+n_s} - \delta^{n_s-1} > 1 - \delta$

4. $n^* + n_s \leq \left(\text{Info}\left(1 - \frac{1}{\pi}\cos^{-1}\left(\frac{2-2c}{2(1-c)+\lambda}\right)\right)\right)^{-1}$

**Proof (sketch)** To prove sample complexity, we first prove that entire partitions will be eliminated if their constituent items accumulate a certain number of losses. We then derive a maximum time for elimination of all non-winning partitions. To prove correctness, we show that conditions 1 and 2 imply the optimality of all winning partition items with respect to other items and prove that the winning partition will be the last remaining partition with high probability. We then use Theorem 1 for the remaining items. The complete proof is found in Appendix E.7 together with a discussion of its implications.

## 8 EXPERIMENTS

**Baselines** We use *Trace-the-Best* (*TTB*) and *Divide-and-Battle* (*DAB*) (Saha & Gopalan, 2019c) as state-of-the-art (to the best of our knowledge) baselines for PAC best-item identification from relative feedback. Due to the lack of competitive and compatble baselines, we consider a modified version of *Dvoretzky–Kiefer–Wolfowitz Tournament* (*DKWT*) (Haddenhorst et al., 2021) as an additional baseline. While *DKWT* does not directly translate to our problem, we argue in Appendix F that *DE* and *DKWT* (with a slight modification) are both able to return a $\epsilon$-optimal Generalized Condorcet winner. We compare both algorithms under this equivalence. A more detailed discussion on baselines is in Appendix G.1.

**Datasets** We consider mainly 3 types of datasets - 1) $N^{16}$: synthetic dataset of 1000 16-dimensional normalized vectors drawn from a multivariate normal distribution, 2) DIM: datasets with 1024 vectors each in well-separated Gaussian clusters in various dimensions from (Fränti et al., 2006) and 3) G2: datasets truncated to 300 vectors in 2 Gaussian clusters with varying degrees of overlap from (Mariescu-Istodor & Zhong, 2016). Notably, these three datasets cover the 3 main scenarios for vector distributions - 1) all vectors are weakly correlated, 2) well formed clusters, 3) most vectors are strongly correlated.

Each setting is run for 100 trials. To increase speed of convergence, we modify the latent scores as follows: $\theta_i = e^{\text{sharpness} \times \mathbf{q} \cdot \mathbf{v}_i}$. We note that this induces faster convergence across all instance optimal algorithms (*DE*, *DEBC*, *DKWT*). We show how sample complexity varies with sharpness in Figure 1. More experimental results can be found in Appendix G.4, including the mean errors ($\frac{1}{2} - p_{i^*1}$) obtained for each experiment.

### 8.1 RESULTS FOR $N^{16}$ DATASET

Figure 1 shows the sample complexities of the various algorithms for the synthetic dataset against varying error bias $\epsilon$, subset size $n_s$ and number of items $n$. *TAB* and *DAB* have sample complexities that are not instance dependent and both are orders of magnitude larger than that of the other baselines.

We note here that *DE* and *DEBC* both find the $\epsilon$-optimal item with at least probability $1 - \delta$ in all the settings. Compared to *DKWT*, both *DE* and *DEBC* outperform it by at least an order of magnitude across all settings. We note that experiments in (Haddenhorst et al., 2021) suggest a similar magnitude for the sample complexity of *DKWT*. The inferred updates are less significant since the random Gaussian vectors are poorly correlated and hence *DEBC* only slightly outperforms *DE*.

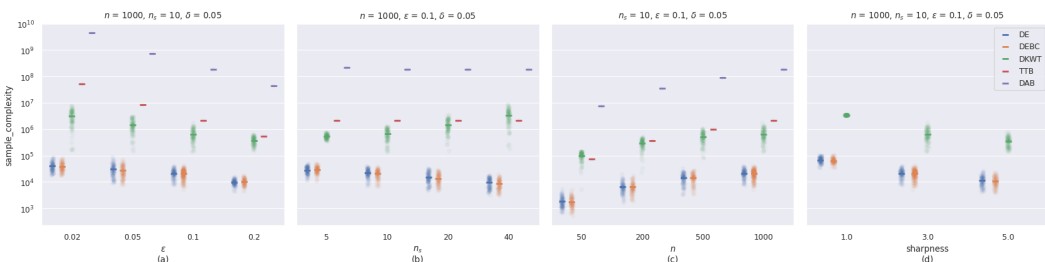

Figure 1: $N^{16}$ dataset: Sample complexities in various settings

Lastly, we note that the general trend of the sample complexity of *DE* and *DEBC* against $n_s$ and $n$ are in agreement with Theorem 1, while sample complexity has a weaker dependence on $\epsilon$ in practice due to dynamic elimination. Notably, their sample complexities scale better against $\epsilon$ compared to *DKWT* which is also designed to be instance optimal and dependent on set hardness.

## 8.2 RESULTS FOR $d = 32$ DIM DATASET

From Figure 2(a), we see that *DE* and *DEBC* still greatly outperform the other baselines in terms of sample complexity. However, we see that for this dataset, *DEBC* has significantly lower sample complexity to *DE* which shows the effectiveness of inferred updates for item clusters. Figure 2(b) and 2(c) show that *DEBC* is robust to perturbations in the item correlation matrix. The increasing sample complexity indicates a reduced reliance on inferred updates as the correlation noise increases, likely because there are fewer significant updates. Figure (d) and (e) show that *DEBC* achieves superior short term performance than *DE*, eliminating more items with a lower running winner error. This indicates that *DEBC* and inferred updates in general can be beneficial in the sample limited setting (Brandt et al., 2022).

## 8.3 RESULTS FOR $d = 32$ G2 DATASET

Figure 3 shows sample complexities against $\epsilon$ for 4 G2 datasets with varying degrees of overlap. The overlap is controlled via the variance of each cluster, where a larger variance leads to larger cluster spread and more overlap between the two clusters. Consequently, we see that *DEBC* has the clearest advantage over *DE* in Figure 3(a) where the degree of overlap is the smallest and inferred updates can most effectively eliminate one of the clusters. Across all datasets, we see that sample complexity is high for *DE*, *DEBC* and *DKWT* due to a half of the item vectors being closely correlated which results in more set plays needed to achieve the required precision for elimination.

## 9 CONCLUSION

In this work, we studied PAC best-item identification from relative feedback. We proposed the *DE* algorithm that flexibly prunes the item set to reserve set plays for potential winning items. We subsequently introduced the notion of *inferred updates*, whereby the win rates of unplayed items

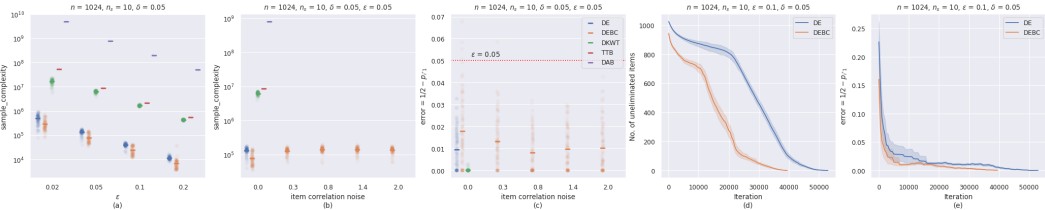

Figure 2: $d = 32$ DIM dataset: (a) sample complexity against $\epsilon$ (b) sample complexity against correlation noise (b) algorithm winner error against correlation noise (d-e) Mean no. of remaining items and error respectively against iteration number

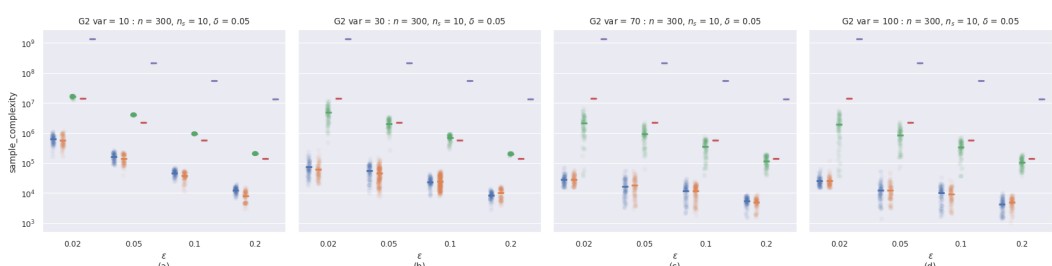

Figure 3: $d = 32$ G2 dataset: Sample complexity against $\epsilon$ across varying degrees of overlap

can be updated through probabilistic Bayesian updates by observing outcomes of sets containing correlated items. We showed that *inferred updates* can be easily incorporated into *DE* to form the *DEBC* algorithm. Experiments show that both *DE* and *DEBC* outperform existing SOTA baselines by a large margin.

This work can be extended in several important directions. First and foremost, while *DE* and *DEBC* clearly exhibit excellent sample complexity performance in practice, this is not reflected in the sample complexity upper bounds. To this end, the theoretical analysis could be extended to instance optimal sample complexity upper bounds. Other interesting directions are the extension of dynamic item elimination to the problem of partial/full ranking with top-$k$ item feedback, as well as the extension of *inferred updates* to the regret minimization problem in multi-duelling bandits (Sui et al., 2017). Additionally, as mentioned in Section 8.2, the superior short term performance of *DEBC* could be beneficial in the sample limited setting. Lastly, it would be interesting and relevant to study how the notion of item similarity can be extended beyond vector correlation to more general settings.

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

# A APPENDIX

# B MORE DETAILS ON INFERRED UPDATES

## B.1 FURTHER DISCUSSION ON COMBINING INFERRED AND EMPIRICAL UPDATES

As mentioned in Section 6.3, jointly considering empirical and inferred updates breaks the identically distributed condition. More precisely, given that $p_{jk}$ is being estimated, empirical updates follow a Bernoulli distribution with mean $p_{jk}$ whereas inferred updates from the conditional probability $p_{jk|ik}$ follow a Bernoulli distribution with mean $p_{ik}$ rescaled according to $\mathrm{pdf}(p_{ik})$ - an approximation for $\mathrm{pdf}(p_{jk})$ given partial information. In fact, we can observe that the predictive posterior distribution is independent of the order in which the updates are applied and view the update sequence in 2 stages - applying all empirical updates in the first stage and inferred updates in the second. Then, each stage is a valid Lévy process, and the posterior distribution from the first stage is supplied as the prior distribution of the second stage.

## B.2 COMBINING INFERRED UPDATES FROM MULTIPLE ITEMS

Consequently, incorporating inferred updates from multiple items can be viewed as a multi-stage update, where each item yields a sequence of iid. updates constituting a single stage. The sequence is independent across all stages - each random variable is only dependent on the underlying distribution it is drawn from. It is trivial to extend Theorem 3 to the multi-item case to show that the sample mean across multiple stages is still an unbiased estimator for $p_{ij}$.

However, in doing so, we are considering the evidence inferred from observations of other item pairs separately instead of jointly, i.e. given $\iota_{ik}$ and $\iota_{hk}$, the inferred updates are derived using the first-order conditional probabilities $p_{jk|ik}$ and $p_{jk|hk}$ instead of $p_{jk|ik \cap hk} = P_\mathbf{q}\left(p_{jk} > \frac{1}{2} \mid p_{hk} > \frac{1}{2} \cap p_{ik} > \frac{1}{2}\right)$. While considering evidence from all item pairs jointly clearly leads to an optimal estimate, computing higher-order probabilities is intractable.

We analyze the feasibility of only considering first-order conditional probabilities via two approaches. Firstly, we derive a lower bound on second order conditional probabilities (Lemma 3) and show that it only deviates slightly from the mean of the constituent first order conditional probabilities when the first order probabilities are close to 1 (Figure 4 (left)). Secondly, for higher order conditional probabilities, we perform Monte Carlo simulations to estimate the average multi-order conditional probability given multiple constituent first-order conditional probabilities (Figure 4 (right)).

Both analyses show that taking the mean of the first order conditional probabilities by treating inferred updates from multiple items independently is a reasonably conservative estimate of the high-order conditional probability when the constituent first order probabilities are sufficiently high. We thus employ the heuristic of weighting the updates by their information content to assign higher importance to probabilities close to 1. Details are found in Appendix B.4.

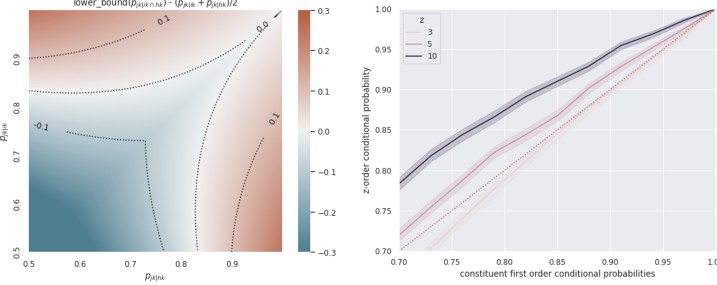

Figure 4: (Left) Deviation of the second order conditional probability lower bound from the mean of the constituent first order conditional probabilities. (Right) Monte Carlo simulation of $z$-order conditional probabilities with 95% confidence interval

### B.3 REGULARIZATION OF CONDITIONAL PROBABILITIES

From Eqn 1, we can see that $p_{jk|ik}$ becomes increasingly sensitive to minor perturbations of $\mathbf{v}_i, \mathbf{v}_j, \mathbf{v}_k$ as $\mathbf{v}_i, \mathbf{v}_j \to \mathbf{v}_k$. Consequently, two vectors that are both $\epsilon$-optimal candidates can yield drastically different conditional probabilities. Intuitively, this sensitivity to slight perturbations leads to unpredictability and poses a problem for its use in a $(\epsilon, \delta)$-PAC algorithm. Particularly, it is prohibitive for formulating of sample complexity lower bounds.

Regularization has been widely used as a way to simplify ill-posed problems in geometry, statistics, and optimization Girosi et al. (1995); Belkin et al. (2005); Bickel et al. (2006). From Appendix E.1, we see that the term $(2\sqrt{(1 - \mathbf{v}_j \cdot \mathbf{v}_k)(1 - \mathbf{v}_i \cdot \mathbf{v}_k)})^{-1}$ comes from $(|\mathbf{v}_i - \mathbf{v}_k||\mathbf{v}_j - \mathbf{v}_k|)^{-1}$ which approaches infinity as $\mathbf{v}_i, \mathbf{v}_j$ approach $\mathbf{v}_k$. Consequently, minor perturbations in the $\mathbf{v}_i \cdot \mathbf{v}_j - \mathbf{v}_i \cdot \mathbf{v}_k - \mathbf{v}_j \cdot \mathbf{v}_k + 1$ are magnified. We add a regularization term to penalize the conditional probabilities when the constituent vectors are too close as follows:

$$p_{jk|ik} = p_{kj|ki} = 1 - \frac{1}{\pi}\cos^{-1}\left(\frac{\mathbf{v}_i \cdot \mathbf{v}_j - \mathbf{v}_i \cdot \mathbf{v}_k - \mathbf{v}_j \cdot \mathbf{v}_k + 1}{2\sqrt{(1 - \mathbf{v}_j \cdot \mathbf{v}_k)(1 - \mathbf{v}_i \cdot \mathbf{v}_k)} + \lambda}\right) \tag{4}$$

where $\lambda$ is the regularization term.

### B.4 ANALYSIS OF HIGH ORDER CONDITIONAL PROBABILITIES

As discussed in Appendix B.2, inferred updates from multiple items is viewed as a multi-stage Bayesian update sequence, and Theorem 3 is used to show the validity of using the sample mean across all stages as an unbiased estimator for $p_{ij}$. We do this instead of jointly considering observations from multiple correlated items because the higher order conditional probabilities are intractable.

Formally, given observed sequences $\iota_{ik}$ and $\iota_{hk}$, the inferred updates are derived using the first-order conditional probabilities $p_{jk|ik}$ and $p_{jk|hk}$ instead of $p_{jk|ik \cap hk} = P_{\mathbf{q}}\left(p_{jk} > \frac{1}{2} \mid p_{hk} > \frac{1}{2} \cap p_{ik} > \frac{1}{2}\right)$. In this section, we will investigate the feasibility of only considering first-order conditional probabilities by a) computing a lower bound on second order conditional probabilities as a function of the constituent first order probabilities and b) performing Monte Carlo simulations to estimate the expected deviation of higher order conditional probabilities from the mean of the constituent first order probabilities.

**Lemma 3 (Lower bound on second order conditional probabilities)** *Given any 4 items* $h, i, j, k \in [n]$, *and assuming WLOG that* $p_{jk|hk} \geq p_{jk|ik}$, *the following is true:*

$$p_{jk|ik \cap hk} \geq 1 - \frac{1 - p_{jk|hk}}{p_{jk|ik}} \tag{5}$$

We visualize the effect of Lemma 3 by plotting the deviation of the lower bound on the second order conditional probability from the mean of the constituent first order conditional probabilities as shown in Figure 4 (right). As can be seen, the worst case deviation is only slightly negative when the first order conditional probabilities are close to 1.

We can extend the formulation to higher order conditional probabilities by considering the intersection of more than 3 hyper hemispherical surfaces. While the exact calculation is intractable, we perform Monte Carlo simulations to estimate the average multi-order conditional probability $p_{jk|\bigcap_i ik}$ given multiple constituent first-order conditional probabilities $p_{jk|ik}$. The details of the simulation are in Appendix C.

The simulation results are shown in Figure 4 (left) which plots the higher order conditional probability against the first order conditional probabilities (we consider a sequence of first order probabilities with equal magnitude) along with the 95% confidence interval. We see that $p_{jk|\bigcap_{i=1}^z ik}$ exhibits a narrow spread, generally increases with $z$, and significantly exceeds the mean of the constituent first order probabilities for $z > 5$. On this basis, we argue that taking the mean of the first order conditional probabilities by treating them inferred updates from multiple items independently is a reasonably conservative estimate of the high-order conditional probability when the constituent

first order probabilities are sufficiently high. We are thus motivated to assign higher importance to first order probabilities that are closer to 1. This is in agreement with the intuition that probabilistic updates that are close to 1 hold more information while probabilistic updates that are close to 0.5 are less significant (e.g. a probabilistic update of 0.5 holds no significance since it is the prior distribution before any updates).

### B.5 Information weighting of inferred updates

To assign higher importance to inferred updates with more certain conditional probabilities, we employ the heuristic of weighting the updates by their information content and modify Eqn. 2 as follows:

$$\iota_{ij}^{full}(t) = \iota_{ij}(t) \cup \iota_{ij}^*(t) \tag{6}$$

$$P_{ij}(t) = \frac{\sum_{p \in \iota_{ij}^{full}(t)} (\text{Info}(p) \times p)}{\sum_{p \in \iota_{ij}^{full}(t)} \text{Info}(p)} \tag{7}$$

where

$$\text{Info}(p) = 1 - (-p \times \log_2 p - (1-p) \times \log_2(1-p)) \tag{8}$$

which is the mutual information content between the update and a $p = 0.5$ prior.

## C   Monte Carlo Simulation of $z$-order conditional probabilities

High order conditional probabilities can be computed as the intersection of more than 3 hyper hemispherical surfaces. While the exact calculation is intractable, we can perform Monte Carlo simulations to estimate the average multi-order conditional probability $p_{jk|\bigcap_i ik}$ given multiple constituent first-order conditional probabilities $p_{jk|ik}$ using the result in Lemma 4. For each simulation, we fix $p_{jk|ik}$ to be of a certain value and compute possible item vectors $i$ that can yield these probabilities. We then randomly initialize query vectors such that they are uniformly distributed on the unit hypersphere according to (Muller, 1959) to estimate $p_{jk|\bigcap_i ik}$.

**Lemma 4 (Generating item vectors subject to conditional probability constraints)** *Given items $j, k$, a random unit vector $\mathbf{r}$ and a desired probability $p$, we can obtain a unit vector $\mathbf{i}$ corresponding to item $i$ such that $p_{jk|ik} = p$ as follows:*

$$\mathbf{v}_{j-k} = \mathbf{v}_j - \mathbf{v}_k, \quad c = \cos((1-p) \times \pi), \quad \mathbf{v}_{j-k}^\perp = \mathbf{r} - (\mathbf{r} \cdot \mathbf{v}_{j-k})\mathbf{v}_{j-k}$$

$$\mathbf{v}_{i-k} = c \times \frac{\mathbf{v}_{j-k}}{|\mathbf{v}_{j-k}|} + \sqrt{1-c^2} \times \frac{\mathbf{v}_{j-k}^\perp}{|\mathbf{v}_{j-k}^\perp|}$$

$$\mathbf{i} = \mathbf{k} - \frac{\mathbf{v}_{i-k}}{2\mathbf{v}_{i-k} \cdot \mathbf{v}_k}$$

**Proof**   It is clear that $|\mathbf{v}_{i-k}| = 1$. Then the following is true:

$$\frac{(\mathbf{v}_j - \mathbf{v}_k) \cdot (\mathbf{v}_i - \mathbf{v}_k)}{|\mathbf{v}_j - \mathbf{v}_k| \times |\mathbf{v}_i - \mathbf{v}_k|} = \frac{\mathbf{v}_{j-k} \cdot \mathbf{v}_{i-k}}{|\mathbf{v}_{j-k}| \times |\mathbf{v}_{i-k}|}$$

$$= \frac{1}{|\mathbf{v}_{j-k}|} \times \frac{c \times \mathbf{v}_{j-k} \cdot \mathbf{v}_{j-k}}{|\mathbf{v}_{j-k}|}$$

$$= c$$

Using the above result, we can complete the proof:

$$p_{jk|ik} = 1 - \frac{1}{\pi}\cos^{-1}\left(\frac{(\mathbf{v}_j - \mathbf{v}_k) \cdot (\mathbf{v}_i - \mathbf{v}_k)}{|\mathbf{v}_j - \mathbf{v}_k| \times |\mathbf{v}_i - \mathbf{v}_k|}\right)$$

$$= 1 - \frac{1}{\pi}\cos^{-1}(c) = p$$

■

For each trial, we assume WLOG that $\mathbf{v}_k = (1, 0, 0, ..., 0)$ and randomly initialize $\mathbf{v}_j$. We can make use of Lemma 4 to obtain a set of $z$ items $\mathcal{V}_i$ such that $p_{jk|ik} = p$ for some $p \in [0.5, 1]$. We then randomly initialize a set of query vectors $\mathcal{V}_q$ that are uniformly distributed on the unit hypersphere by initializing $d$-dimensional Gaussian random vectors and normalizing them (Muller, 1959). We can then estimate $p_{jk|\bigcap_i ik}$ by computing the ratio:

$$\frac{|\{\mathbf{q} \in \mathcal{V}_q : \mathbf{q} \cdot \mathbf{v} > \mathbf{q} \cdot \mathbf{v}_k \ \ \forall v \in \mathcal{V}_i \cap \{\mathbf{v}_j\}\}|}{|\{\mathbf{q} \in \mathcal{V}_q : \mathbf{q} \cdot \mathbf{v} > \mathbf{q} \cdot \mathbf{v}_k \ \ \forall v \in \mathcal{V}_i|}$$

For each pair of $(z, p)$ data point, we perform 4000 trials. The number of query vectors $|\mathcal{V}_q|$ is set to $1 \times 10^5$. $d$ is set as 32.

## D  ALGORITHMS

### D.1  DYNAMIC ELIMINATION (DE)

The complete algorithm is given as Algorithm 1 with a subroutine given in Algorithm 2 for updating of the played set in response to the user feedback which we restate here for completeness' sake.

**Remarks**  The algorithm draws inspiration from *Trace-the-Best* in (Saha & Gopalan, 2019c) and maintains a prevailing winner that we term the *running winner* that is at least pairwise $\epsilon$-optimal to all items that have been played so far. Each item pair is played for the required number of times to establish the winner with sufficient certainty before it is removed permanently. However, *Trace-the-Best* removes an entire set (except the winner) only when the set winner is established instead of removing items once they are no longer potential winners. We improve on this and implement flexible item elimination while achieving an improved worst case sample complexity. A crucial component of this is running winner inheritance in which the incoming running winner inherits the pairwise interactions of the outgoing winner during running winner replacement. Additionally, while *DE* is a superior algorithm in its own right as we show in Section 8, its ability to dynamically eliminate items facilitates straightforward accommodation of inferred updates. Firstly, without dynamic item elimination, inferred updates can only be eliminated outside of set plays. Otherwise, once items are added into a set, their previously accumulated inferred updates are redundant since they can only be removed together with other items in the set. Secondly, the importance weighting of inferred updates means that more inferred updates are required for item elimination. This means that a running winner can potentially be replaced before the items that have accumulated inferred updates from it have been eliminated. Consequently, these updates are redundant since those items will have to accumulate updates with the new running winner. Running winner inheritance effectively solves this problem with theoretical correctness guarantees.

### D.2  DYNAMIC ELIMINATION BY CORRELATION (DEBC)

The complete algorithm is given as Algorithm 3 with the set update subroutine given in Algorithm 4. While it is largely similar to *DE*, we have included it here in full for completeness' sake. The areas where it differs from *DE* are highlighted in red.

**Remarks**  Compared to *DE*, *DEBC* leverages the correlation matrix in two areas - item selection and inferred updates. Firstly, the correlation matrix is used to select items that are least correlated with items that have been played to rapidly sweep the item space and increase the probability of playing an item close to the optimal item is high which improves regret performance in the short term. Secondly, it is used to implement inferred updates to the preference matrix for item pairs that have not been played. This maximizes the information gain from each iteration.

---

**Algorithm 1:** Dynamic Elimination (*DE*)

**Input:** set of items: $[n]$, subset size: $n_s$, error bias: $\epsilon > 0$, confidence parameter: $\delta > 0$

**Initialize:** uneliminated item set: $S \leftarrow [n]$, item subset to play: $G \leftarrow \emptyset$, empirical pairwise win

   ratio matrix: $\mathbf{W} \leftarrow [0]^{n \times n}$, $\gamma \leftarrow \left\lceil \frac{n}{n_s} \right\rceil$, $m \leftarrow \frac{2 \ln(\gamma/\delta)}{\epsilon^2}$

**1** **while** $|S| > 1$ **do**

**2**    **if** $|G| < n_s$ **then**

**3**      $a \leftarrow$ random item from $S \backslash G$ // randomly select unplayed item

**4**      $G \leftarrow G \cup \{a\}$ // build initial item subset/replenish eliminated item

**5**    **if** $|G| = n_s$ **then**

**6**      Play set $G$, $i \leftarrow$ winning item

**7**      $\forall k \in G, k \neq i : W_{ik} \leftarrow W_{ik} + 1$ // Update empirical pairwise win ratios

**8**      $\mathbf{N} \leftarrow \mathbf{W} + \mathbf{W}^T, \ \ \mathbf{P} = \mathbf{W}/\mathbf{N}$

**9**      $\mathbf{U} = \mathbf{P} + \sqrt{\frac{\ln(\gamma/\delta)}{2\mathbf{N}}}$ // Update upper confidence bound matrix

        // run *update-set* to eliminate items, update running winner

**10**      $G, S, i^* \leftarrow$ *update-set*$(G, i^*, \mathbf{U}, \mathbf{P}, \mathbf{N}, S, m, \epsilon)$

        // keep only potential Condorcet winners

**11**      $S \leftarrow \{j \in S : \min_{j' \in S} U_{jj'} \geq \frac{1}{2}\}$

**12**      $S \leftarrow S \backslash \{j \in S : P_{i^*j} \geq \frac{1}{2} - \frac{\epsilon}{2} \text{ and } N_{i^*j} \geq m\}$

---

**Algorithm 2:** *DE update-set* subroutine - eliminates suboptimal items, updates item subset and running winner

**Input:** subset $G$, current winner $i^*$, upper confidence bound matrix $\mathbf{U}$, preference matrix $\mathbf{P}$,

   count matrix $\mathbf{N}$, potential candidate set: $S$, max no. of updates $m$, error bias $\epsilon$

**Initialize:** updated subset $H \leftarrow \emptyset$, potential running winner challengers

   $W \leftarrow \{j \in G \backslash \{i^*\} : N_{i^*j} \geq m, P_{i^*j} < \frac{1}{2} - \frac{\epsilon}{2}\}$

**1** **for** $j \in G \backslash (\{i^*\} \cup W)$ **do**

**2**    **if** $U_{ji^*} < 1/2$ *or* $N_{i^*j} \geq m$ **then**

       // eliminate item if it is not a potential Condorcet winner

**3**      $S \leftarrow S \backslash \{j\}$

**4**      $a \leftarrow$ random item from $S \backslash G$

**5**      $H \leftarrow H \cup \{a\}$ // replace with randomly selected item

**6**    **else**

**7**      $H \leftarrow H \cup \{j\}$

   // update current running winner $i^*$ with new running winner $i$

**8** **if** $|W| \neq 0$ **then**

**9**    $i \leftarrow \arg\max_{j \in W} P_{i^*j}$ // item with highest win prob. over current winner $i^*$

   // the incoming running winner inherits the win/losses from the

     outgoing winner as a conservative estimate

**10**    $\forall j \in S \backslash \{i\} : P_{ij} \leftarrow P_{i^*j} \times N_{i^*j} + P_{ij} \times N_{ij}, \ \ N_ij \leftarrow N_{ij} + N_{i^*j} \ i^* \leftarrow i$

**11**    $H \leftarrow H \cup W$

**12** **else**

**13**    $H \leftarrow H \cup \{i^*\}, i \leftarrow i^*$

**Output:** $H, S, i$

---

**Algorithm 3:** Dynamic Elimination By Corelation (*DEBC*)

---

**Input:** set of items: $[n]$, subset size: $n_s$, error bias: $\epsilon > 0$, confidence parameter: $\delta > 0$, item correlation matrix: **C**, conditional probability regularization term: $\lambda > 0$

**Initialize:** $S \leftarrow [n], G \leftarrow \emptyset, \mathbf{W} \leftarrow [0]^{n \times n}, \gamma \leftarrow \left\lceil \frac{n}{n_s} \right\rceil, m \leftarrow \frac{2 \ln(\gamma/\delta)}{\epsilon^2}$

**1 while** $|S| > 1$ **do**

**2**    **if** $|G| = 0$ **then**

**3**      $a \leftarrow \underset{i \in S}{\arg\max} \sum_{j \in S} C_{ij}$ // item most correlated with other items

**4**      $i^* \leftarrow a$

**5**    **else if** $|G| < n_s$ **then**

**6**      $a \leftarrow \underset{i \in S \setminus G}{\arg\min} \left( \underset{j \in G}{\max} C_{ij} \right)$ // item uncorrelated with existing items in $G$

**7**    $G \leftarrow G \cup \{a\}$

**8**    **if** $|G| = n_s$ **then**

**9**      Play set $G, i \leftarrow$ winning item

**10**      $\forall k \in G, k \neq i : W_{ik} \leftarrow W_{ik} + 1$ // Empirical updates

**11**      $\forall k \in G, \forall j \in S, k \neq i :$ // Inferred updates

**12**        $\rho \leftarrow \text{Info}(p_{jk|ik})$

**13**        $W_{jk} \leftarrow W_{jk} + \rho \times p_{jk|ik}, \quad W_{kj} \leftarrow W_{kj} + \rho \times (1 - p_{jk|ik})$

**14**        $\rho \leftarrow \text{Info}(p_{ij|ik})$

**15**        $W_{ij} \leftarrow W_{ij} + \rho \times p_{ij|ik}, \quad W_{ji} \leftarrow W_{ji} + \rho \times (1 - p_{ij|ik})$

**16**      $\mathbf{N} \leftarrow \mathbf{W} + \mathbf{W}^T, \quad \mathbf{P} = \mathbf{W}/\mathbf{N}, \quad \mathbf{U} = \mathbf{P} + \sqrt{\frac{\ln(\gamma/\delta)}{2\mathbf{N}}}$

**17**      $G, S, i^* \leftarrow$ *update-set*$(G, i^*, \mathbf{U}, \mathbf{P}, \mathbf{N}, S, m, \epsilon, \mathbf{C})$
       // keep only potential Condorcet winners

**18**      $S \leftarrow \{j \in S : \underset{j' \in S}{\min} U_{jj'} \geq \frac{1}{2}\}$

**19**      $S \leftarrow S \setminus \{j \in S : P_{i^*j} \geq \frac{1}{2} - \frac{\epsilon}{2} \text{ and } N_{i^*j} \geq m\}$

---

# E PROOFS

## E.1 PROOF OF THEOREM 2

**Theorem 2 (Conditional probabilities of win ratios)** *Given items $i, j, k \in [n]$, the following holds true:*

$$p_{jk|ik} = p_{kj|ki} = 1 - \frac{1}{\pi} \cos^{-1} \left( \frac{\mathbf{v}_i \cdot \mathbf{v}_j - \mathbf{v}_i \cdot \mathbf{v}_k - \mathbf{v}_j \cdot \mathbf{v}_k + 1}{2\sqrt{(1 - \mathbf{v}_j \cdot \mathbf{v}_k)(1 - \mathbf{v}_i \cdot \mathbf{v}_k)}} \right) \quad (1)$$

**Proof** We begin by stating and proving the following lemma:

**Lemma 5** *Given a fixed pair of unit vectors $\mathbf{v}_i, \mathbf{v}_j \in \mathbb{R}^d$, for any vector $\mathbf{q} \in \mathbb{R}^d$ that ends on the $d$-dimensional unit hyperspherical cap with axis $\mathbf{v}_i - \mathbf{v}_j$ and colatitude angle $\pi/2$, $\mathbf{q} \cdot \mathbf{v}_i \geq q \cdot \mathbf{v}_j$ must be true.*

**Proof of Lemma 5** Note that the colatitude angle is the largest angle formed by the axis and a vector on the hyperspherical cap. As such, we have

$$0 \leq \mathbf{q} \cdot (\mathbf{v}_i - \mathbf{v}_j) = \mathbf{q} \cdot \mathbf{v}_i - \mathbf{q} \cdot \mathbf{v}_j \Rightarrow \mathbf{q} \cdot \mathbf{v}_i \geq \mathbf{q} \cdot \mathbf{v}_j$$

∎

Let $\text{Cap}(\phi, \mathbf{x})$ denote the hyperspherical cap with colatitude angle $\phi$ and axis $x \in \mathbb{R}^d$, $\text{Area}(...)$ denote the area of the input region and $\text{Cap}_1 \cap \text{Cap}_2$ denote the intersection of two caps.

$$p_{jk|ik} = \frac{Pr(\mathbf{q} \cdot \mathbf{v}_j > \mathbf{q} \cdot \mathbf{v}_k \cap \mathbf{q} \cdot \mathbf{v}_i > \mathbf{q} \cdot \mathbf{v}_k)}{Pr(\mathbf{q} \cdot \mathbf{v}_i > \mathbf{q} \cdot \mathbf{v}_k)}$$

---

**Algorithm 4:** *DEBC update-set* subroutine

---

**Input:** subset $G$, current winner $i^*$, upper confidence bound matrix $\mathbf{U}$, preference matrix $\mathbf{P}$,
count matrix $\mathbf{N}$, potential candidate set: $S$, item correlation matrix: $\mathbf{C}$, max no. of
updates $m$, error bias $\epsilon$

**Initialize:** $H \leftarrow \emptyset, W \leftarrow \{j \in G \backslash \{i^*\} : N_{i^*j} \geq m, P_{i^*j} < \frac{1}{2} - \frac{\epsilon}{2}\}$

**1 for** $j \in G \backslash (\{i^*\} \cup W)$ **do**

    // keep only potential Condorcet winners

**2**    **if** $U_{ji^*} < 1/2$ ***or*** $N_{i^*j} \geq m$ **then**

**3**        $S \leftarrow S \backslash \{j\}$

        // replace with item uncorrelated with items that have been
played before

**4**        $H \leftarrow H \cup \underset{j \in S \backslash G}{\arg\min} \left( \underset{k \in ([n] \backslash S) \cap G}{\max} C_{jk} \right)$

**5**    **else**

**6**        $H \leftarrow H \cup \{j\}$

**7 if** $|W| \neq 0$ **then**

**8**    $i \leftarrow \underset{j \in W}{\arg\max} P_{i^*j}$ // potential replacement for running winner

    // the incoming running winner inherits the win/losses from the
outgoing winner as a conservative estimate

**9**    $\forall j \in S \backslash \{i\} : P_{ij} \leftarrow P_{i^*j} \times N_{i^*j} + P_{ij} \times N_{ij}, \ N_{ij} \leftarrow N_{ij} + N_{i^*j} \ i^* \leftarrow i$

**10**    $H \leftarrow H \cup W$

**11 else**

**12**    $H \leftarrow H \cup \{i^*\}$

**13** $G \leftarrow H$

**Output:** $G, S, i^*$

---

$$\overset{(a)}{=} \frac{\text{Area}(\text{Cap}(\pi/2, \mathbf{v}_j - \mathbf{v}_k) \cap \text{Cap}(\pi/2, \mathbf{v}_i - \mathbf{v}_k))}{\text{Area}(\text{Cap}(\pi/2, \mathbf{v}_i - \mathbf{v}_k))}$$

$$\overset{(b)}{=} 1 - \frac{\Delta_\phi(\mathbf{v}_j - \mathbf{v}_k, \mathbf{v}_i - \mathbf{v}_k)}{\pi}$$

$$= 1 - \frac{1}{\pi}\cos^{-1}\left(\frac{(\mathbf{v}_j - \mathbf{v}_k) \cdot (\mathbf{v}_i - \mathbf{v}_k)}{|\mathbf{v}_j - \mathbf{v}_k| \times |\mathbf{v}_i - \mathbf{v}_k|}\right)$$

$$= 1 - \frac{1}{\pi}\cos^{-1}\left(\frac{\mathbf{v}_i \cdot \mathbf{v}_j - \mathbf{v}_i \cdot \mathbf{v}_k - \mathbf{v}_j \cdot \mathbf{v}_k + 1}{2\sqrt{(1 - \mathbf{v}_j \cdot \mathbf{v}_k)(1 - \mathbf{v}_i \cdot \mathbf{v}_k)}}\right)$$

where $\Delta_\phi(..., ...)$ returns the angle between two vectors. We use Lemma 5 for equality (a) while equality (b) holds when we observe that the intersection between the two hyper-hemispherical caps is a hyperspherical wedge with dihedral angle $\pi - \Delta_\phi(\mathbf{v}_j - \mathbf{v}_k, \mathbf{v}_i - \mathbf{v}_k)$. The second equality in Theorem 2 is proven in a similar manner. We include it below for completeness' sake.

$$p_{kj|ki} = \frac{Pr(\mathbf{q} \cdot \mathbf{v}_k > \mathbf{q} \cdot \mathbf{v}_k \cap \mathbf{q} \cdot \mathbf{v}_k > \mathbf{q} \cdot \mathbf{v}_i)}{Pr(\mathbf{q} \cdot \mathbf{v}_k > \mathbf{q} \cdot \mathbf{v}_i)}$$

$$= \frac{\text{Area}(\text{Cap}(\pi/2, \mathbf{v}_k - \mathbf{v}_j) \cap \text{Cap}(\pi/2, \mathbf{v}_k - \mathbf{v}_i))}{\text{Area}(\text{Cap}(\pi/2, \mathbf{v}_k - \mathbf{v}_i))}$$

$$= 1 - \frac{\Delta_\phi(\mathbf{v}_k - \mathbf{v}_j, \mathbf{v}_k - \mathbf{v}_i)}{\pi}$$

$$= 1 - \frac{\Delta_\phi(\mathbf{v}_j - \mathbf{v}_k, \mathbf{v}_i - \mathbf{v}_k)}{\pi}$$

$$= 1 - \frac{1}{\pi}\cos^{-1}\left(\frac{\mathbf{v}_i \cdot \mathbf{v}_j - \mathbf{v}_i \cdot \mathbf{v}_k - \mathbf{v}_j \cdot \mathbf{v}_k + 1}{2\sqrt{(1 - \mathbf{v}_j \cdot \mathbf{v}_k)(1 - \mathbf{v}_i \cdot \mathbf{v}_k)}}\right)$$

■

### E.2 PROOF OF THEOREM 3

**Theorem 3 (Estimating $p_{ij}$ from inferred updates)** *For any item pair $i, j$, given a sequence of binary empirical updates $\iota_{ij}(t)$ and a sequence of inferred updates $\iota_{ij}^*(t)$, the sample mean*

$$P_{ij}(t) = \frac{1}{|\iota_{ij}(t)|} \sum_{x \in \iota_{ij}(t)} x + \frac{1}{|\iota_{ij}^*(t)|} \sum_{p \in \iota_{ij}^*(t)} p \qquad (2)$$

*is an unbiased estimator of $p_{ij}$.*

**Proof** We begin by proving the following lemma :

**Lemma 6 (Probabilistic Bayesian updates to mixtures of beta distributions)** *Let X be a random variable whose probability is given by a sum of Beta distributions, i.e.*

$$\text{pdf}(X) = \sum_{i=0}^{i=N-1} c_i \text{Beta}(\alpha_i, \beta_i)$$

$$\forall i \in [0, N-1] : \alpha_i + \beta_i = \eta$$

$$\sum_{i=0}^{i=N-1} c_i = 1$$

*Then, the following is true:*

$$\text{pdf}(X | Pr(Y \text{ Bernoulli}(X) = 1) = p) = \sum_{i=0}^{i=2N-1} d_i \text{Beta}(\alpha_i', \beta_i')$$

$$\forall i \in [0, 2N-1] : \alpha_i' + \beta_i' = \eta + 1$$

$$\sum_{i=0}^{i=2N-1} d_i = 1$$

*and the mean of the posterior distribution is*

$$\frac{\eta \bar{X} + p}{\eta + 1} \qquad (9)$$

*where $\bar{X}$ denotes the mean value of $X$.*

**Proof** Using Jeffrey Conditionalization, we have

$$\text{pdf}(X | Pr(Y \sim \text{Bernoulli}(X) = 1) = p)$$

$$= p \times \sum_{i=0}^{i=N-1} c_i \text{Beta}(\alpha_i + 1, \beta_i) + (1-p) \times \sum_{i=0}^{i=N-1} c_i \text{Beta}(\alpha_i, \beta_i + 1)$$

$$= \sum_{i=0}^{i=N-1} c_i \left( p \times \text{Beta}(\alpha_i + 1, \beta_i) + (1-p) \times \text{Beta}(\alpha_i, \beta_i + 1) \right)$$

$$= \sum_{i=0}^{i=2N-1} d_i \text{Beta}(\alpha_i', \beta_i')$$

where

$$\alpha'_i, \beta'_i = \begin{cases} \alpha_{\frac{i}{2}} + 1, \beta_{\frac{i}{2}} & \text{if } i \text{ is even} \\ \alpha_{\lfloor \frac{i}{2} \rfloor}, \beta_{\lfloor \frac{i}{2} \rfloor} + 1 & \text{if } i \text{ is odd} \end{cases}$$

$$d_i = \begin{cases} c_{i/2} \times p & \text{if } i \text{ is even} \\ c_{\lfloor \frac{i}{2} \rfloor} \times (1 - p) & \text{if } i \text{ is odd} \end{cases}$$

Consequently, it is clear that $\forall i \in [0, 2N-1] : \alpha'_i + \beta'_i = \eta + 1$ and $\sum_{i=0}^{i=2N-1} d_i = 1$. Denoting the mean of the conditional probability distribution by $\bar{X}^*$, we have

$$\bar{X} = \sum_{i=0}^{i=N-1} \frac{c_i \alpha_i}{\eta}$$

$$\bar{X}^* = \sum_{i=0}^{i=2N-1} \frac{d_i \alpha'_i}{\eta + 1}$$

$$= \sum_{i=0}^{i=N-1} p \times \frac{c_i(\alpha_i + 1)}{\eta + 1} + \sum_{i=0}^{i=N-1} (1 - p) \times \frac{c_i \alpha_i}{\eta}$$

$$= \sum_{i=0}^{i=N-1} \frac{c_i \alpha_i + p}{\eta + 1}$$

$$= \frac{\eta}{\eta + 1} \sum_{i=0}^{i=N-1} \frac{c_i \alpha_i + p}{\eta}$$

$$= \frac{\eta}{\eta + 1} \left( \bar{X} + p/\eta \right)$$

$$= \frac{\eta \bar{X} + p}{\eta + 1}$$

∎

It is instructive to assume a Bayes prior $\text{Beta}(1, 1)$ (uniform) for $p_{ij}$ before any updates are applied. Empirical updates can be treated as probabilistic updates with $p = 1$. We can thus consider a single sequence of probabilistic updates $\iota_{ij}^{full}(t) = \iota_{ij}(t) \cup \iota_{ij}^*(t)$. By applying Lemma 4 iteratively , we have that the resulting predictive posterior distribution is also a mixture of Beta distributions that constitutes a valid probability distribution (normalized and continuous).

We now aim to show that the mean of this distribution is indeed the sample mean. We denote the mean of the predictive posterior distribution after $m$ updates as $t$ as $\mu(m)$. Since we start with a uniform prior distribution, we have $\mu(0) = 0.5$. Denoting the $i^{th}$ element of $\iota_{ij}^{full}(t)$ as $x_i$ We can prove that $\mu(m) = \frac{1}{m} \sum_{i=1}^{m} x_i$ by mathematical induction:

Let $Q(m)$ denote the proposition that $\mu(m) = \sum_{i=0}^{m} x_i$ for all $m \in \mathbb{N}$, i.e. the sample mean is the posterior distribution mean. Since $\mu(m) = \frac{0 \times 0 + x_1}{0 + 1} = x_1$, $Q(1)$ is true. We want to show $Q(m)$ is true $\Rightarrow Q(m+1)$ is true.

$$Q(m) \Rightarrow \mu(m) = \frac{1}{m} \sum_{i=1}^{m} x_i$$

$$\Rightarrow \mu(m+1) = \frac{1}{m+1} \left( x_{m+1} + \sum_{i=1}^{m} x_i \right) = \frac{1}{m+1} \sum_{i=1}^{m+1} x_i$$

$$\Rightarrow Q(m+1)$$

By mathematical induction, $Q(m)$ true for all $m \in \mathbb{N}$. The proof of Theorem 3 is thus complete. ∎

### E.3 PROOF OF LEMMA 3

**Lemma 3 (Lower bound on second order conditional probabilities)** *Given any 4 items $h, i, j, k \in [n]$, and assuming WLOG that $p_{jk|hk} \geq p_{jk|ik}$, the following is true:*

$$p_{jk|ik \cap hk} \geq 1 - \frac{1 - p_{jk|hk}}{p_{jk|ik}} \tag{5}$$

**Proof** We begin by proving the following Lemma:

**Lemma 7 (Lower bound on intersection of 3 regions)** *Let $A$, $B$ and $C$ denote regions on some arbitrary surface such that $A$ and $B$ have area $a$. Let the area of some region $R$ be given by $r_R \times a$ (then $r_A = r_B = 1$). Given that $a_C = ra$, we have*

$$\frac{r_{A \cap B \cap C}}{r_{B \cap C}} \geq \frac{r_{A \cap C} a_C + r_{A \cap B} - 1}{r_{A \cap C} r_C + r_{A \cap B} - 1 + \min(1 - r_{A \cap B}, r_C - r_C r_{A \cap C})}$$

**Proof**

$$
\begin{aligned}
0 &\leq \text{Area}(A \cap (\neg B) \cap (\neg C)) \\
&= a - (r_{A \cap B} a - r_{A \cap B \cap C} a) - (r_{A \cap C} r_C a - r_{A \cap B \cap C} a) - r_{A \cap B \cap C} a \\
&= a(1 - r_{A \cap B} - r_{A \cap C} r_C + r_{A \cap B \cap C}) \\
\Rightarrow \quad & r_{A \cap B \cap C} \leq r_{A \cap C} r_C + r_{A \cap B} - 1
\end{aligned}
$$

And

$$
\begin{aligned}
r_{(\neg A) \cap B \cap C} a &\leq \min(a - r_{A \cap B} a, a - r_{A \cap B} r_C a) \\
\Rightarrow \quad r_{(\neg A) \cap B \cap C} &\leq \min(1 - r_{A \cap B}, 1 - r_{A \cap B} r_C)
\end{aligned}
$$

Consequently,

$$
\begin{aligned}
\frac{r_{A \cap B \cap C}}{r_{B \cap C}} &= \frac{r_{A \cap B \cap C}}{r_{A \cap B \cap C} + r_{(\neg A)} \cap B \cap C} \\
&\geq \frac{r_{A \cap C} a_C + r_{A \cap B} - 1}{r_{A \cap C} r_C + r_{A \cap B} - 1 + \min(1 - r_{A \cap B}, r_C - r_C r_{A \cap C})}
\end{aligned}
$$

which completes the proof of Lemma 7. ∎

From Lemma 5, the query vectors $q$ that satisfy $p_{ij} > 1/2$ for any $i, j \in [n]$ end of the surface of a hyper-hemispherical cap. We can thus interpret the second-order conditional probability as a ratio of the intersection areas of hyper-hemispherical caps. Applying Lemma 7, we have

$$p_{jk|ik \cap hk} \geq \frac{p_{jk|hk} + p_{jk|ik} - 1}{p_{jk|ik}} = 1 - \frac{1 - p_{jk|hk}}{p_{jk|ik}} \tag{10}$$

which completes the proof. ∎

### E.4 PROOF OF THEOREM 1

**Theorem 1 (Sample complexity and correctness of *DE* in the general case)** *DE is $(\epsilon, \delta)$-PAC with worst-case sample complexity $O(\frac{n}{\epsilon^2} \ln(\frac{n}{n_s \delta}))$.*

#### E.4.1 PROOF OF CORRECTNESS

We first prove the correctness of the algorithm. Let us recall that the algorithm should output an $\epsilon$-optimal item $i^*$ (i.e. $p_{i^*1} > \frac{1}{2} - \epsilon$, where $1$ is the actual Condorcet winner). We first state the following Lemma:

**Lemma 8 (Hoeffding's Inequality)** *For any item pair $i, j \in [n]$ and $\delta, \epsilon > 0$, given a sequence of $N$ updates $\iota_{ij}(t)$ such that $N \geq \frac{-\ln(\delta)}{2\epsilon^2}$, the sample mean $P_{ij}(t)$ is bounded as follows:*

$$Pr(|p_{ij} - P_{ij}(t)| \geq \epsilon) \leq \delta \tag{11}$$

**Proof**  From Theorem 3, we have that the sample mean of the update sequence is an unbiased estimator of $p_{ij}$. From Section B.2, we also have that the updates are independent (though not identically distributed when inferred updates are considered). This allows us to apply the Hoeffding's Inequality (Hoeffding, 1994; Saha & Gopalan, 2019c) as follows:

$$Pr(|p_{ij} - P_{ij}(t)| \geq \eta/N) \leq \exp\left(-\frac{2\eta^2}{N}\right)$$

Substituting $\delta = \exp\left(-\frac{2\eta^2}{N}\right)$ and $\epsilon = \frac{\eta}{N}$ yields the expression in Eqn. 11. ∎

**Notation**  We then aim to prove the correctness of the running winner in the *DE* algorithm. To do so, we first define some notation: Let the time step $t$ denote the number of sets played since the beginning of the algorithm. For any variable $x$ that changes with $t$, let $x(t)$ denote the value of the variable at the start of time step $t$ unless otherwise stated. Let $Q(t) = [n]\backslash S(t)$ denote the set of eliminated items at time step $t$ since the beginning and $R(t) = Q(t+1)\backslash Q(t)$ denote the set of items eliminated during time step $t$.

**Lemma 9 (Running winner update in *DE*)**  *Given that at some time step $t \geq 0$, $i^*(t+1) \neq i^*(t)$, i.e. the running winner is replaced. Then, the following must be true:*

$$Pr\left(p_{i^*(t+1)i^*(t)} > \frac{1}{2}\right) > 1 - \frac{\delta}{\gamma} \tag{12}$$

**Proof**  We have that $i^*(t+1) \neq i^*(t)$ iff. $N_{i^*(t)j} \geq m$, $P_{i^*(t)i^*(t+1)} < \frac{1}{2} - \frac{\epsilon}{2} \Rightarrow P_{i^*(t+1)i^*(t)} \geq \frac{1}{2} + \frac{\epsilon}{2}$. Applying Lemma 8, we have:

$$Pr\left(\left(\frac{1}{2} + \frac{\epsilon}{2} - p_{i^*(t+1)i^*(t)}\right) \geq \frac{\epsilon}{2}\right)$$

$$\leq Pr\left((P_{i^*(t+1)i^*(t)}(t) - p_{i^*(t+1)i^*(t)}) \geq \frac{\epsilon}{2}\right) \leq \frac{\delta}{\gamma}$$

$$\Rightarrow Pr\left(p_{i^*(t+1)i^*(t)} \leq \frac{1}{2}\right) \leq \frac{\delta}{\gamma}$$

$$\Rightarrow Pr\left(p_{i^*(t+1)i^*(t)} > \frac{1}{2}\right) > 1 - \frac{\delta}{\gamma}$$

**Lemma 10 (Running winner inheritance)**  *Given that at some time step $t \geq 0$, $i^*(t+1) \neq i^*(t)$, i.e. the running winner is replaced, the following must be true for any item $j$:*

$$Pr\left(p_{i^*(t+1)j} > p_{i^*(t)j}\right) > 1 - \frac{\delta}{\gamma} \tag{13}$$

**Proof**

$$Pr\left(p_{i^*(t+1)j} > p_{i^*(t)j}\right) = Pr\left(\frac{\theta_{i^*(t+1)}}{\theta_{i^*(t+1)} + \theta_j} > \frac{\theta_{i^*(t)}}{\theta_{i^*(t)} + \theta_j}\right)$$

$$= Pr(\theta_{i^*(t+1)} > \theta_{i^*(t)})$$

$$= Pr\left(p_{i^*(t+1)i^*(t)} > \frac{1}{2}\right) > 1 - \frac{\delta}{\gamma}$$

where we have used Lemma 9 in the last inequality ∎

**Lemma 11 (Validity of inherited $P_{ij}$)** *Let us denote a sequence of $K$ running winners $\{i_1^*, i_2^*, ... i_K^*\}$ ordered by increasing time step. Let $P_{i_\kappa^* j}$ be the sample estimate given some item $j$ corresponding to $n_\kappa$ samples such that*

$$\forall \kappa \in 1, 2, ... K : Pr\left((P_{i_\kappa^*} - p_{i_\kappa^*}) < \epsilon\right) > 1 - \exp\left(-2n_\kappa \epsilon^2\right)$$

*where $n_\kappa = n_{i_\kappa^* | \{i_\kappa^*, j\}} + n_{j | \{i_\kappa^*, j\}}$ denotes the number of times either $i_\kappa^*$ or $j$ wins a set. Then, given*

$$P_{i_K^* j}^{\text{inh}} = \frac{1}{n_{1,K}} \sum_{\kappa=0}^{K} n_\kappa P_{i_\kappa^*}$$

$$n_{\kappa_0, \delta_\kappa} = \sum_{\kappa=\kappa_0}^{\kappa_0 + \delta_\kappa - 1} n_\kappa$$

*we have*

$$Pr((P_{i_K^* j}^{\text{inh}} - p_{i_K^* j}) < \epsilon) > (1 - \exp\left(-2n_{1,K}\epsilon^2\right)) \times \left(1 - \frac{\delta(K-1)}{\gamma}\right)$$

**Proof** We first consider the case with 2 running winners $i_\kappa^*, i_{\kappa+1}^*$, and :

$$Pr\left(\left(\frac{n_\kappa P_{i_\kappa^* j} + n_{\kappa+1} P_{i_{\kappa+1}^* j}}{n_{\kappa,2}} - p_{i_{\kappa+1}^* j}\right) < \frac{n_\kappa \epsilon + n_{\kappa+1} \epsilon}{n_{\kappa,2}}\right)$$

$$= Pr\left(\left(\frac{n_\kappa P_{i_\kappa^* j}}{n_{\kappa,2}} - \frac{n_\kappa p_{i_{\kappa+1}^* j}}{n_{\kappa,2}}\right) + \left(\frac{n_{\kappa+1} P_{i_{\kappa+1}^* j}}{n_{\kappa,2}} - \frac{n_{\kappa+1} p_{i_{\kappa+1}^* j}}{n_{\kappa,2}}\right)\right.$$

$$\left. < \frac{n_\kappa \epsilon}{n_{\kappa,2}} + \frac{n_{\kappa+1} \epsilon}{n_{\kappa,2}}\right)$$

$$\geq Pr\left(\left(\frac{n_\kappa P_{i_\kappa^* j}}{n_{\kappa,2}} - \frac{n_\kappa p_{i_\kappa^* j}}{n_{\kappa,2}}\right) + \left(\frac{n_{\kappa+1} P_{i_{\kappa+1}^* j}}{n_{\kappa,2}} - \frac{n_{\kappa+1} p_{i_{\kappa+1}^* j}}{n_{\kappa,2}}\right)\right.$$

$$\left. < \frac{n_\kappa \epsilon}{n_{\kappa,2}} + \frac{n_{\kappa+1} \epsilon}{n_{\kappa,2}}\right) \times Pr\left(p_{i_{\kappa+1}^* j} > p_{i_\kappa^* j}\right)$$

$$\overset{\text{(a)}}{>} (1 - \exp\left(-2n_{\kappa,2}\epsilon^2\right)) \times \left(1 - \frac{\delta}{\gamma}\right)$$

where for inequality (a) we have used Lemma 8 for the first term and Lemma 9 for the second term. For the first term, we note that the expression is the confidence interval of a sequence of independent random variables belonging to two distributions which still meets the conditions for application of Hoeffding's inequality. We can apply this iteratively to obtain

$$Pr\left(\frac{1}{n_{1,K}} \sum_{\kappa=0}^{K} -p_{i_K^* j} < \epsilon\right) = (1 - \exp\left(-2n_{1,K}\epsilon^2\right)) \times \left(1 - \frac{\delta}{\gamma}\right)^{(K-1)}$$

$$> (1 - \exp\left(-2n_{1,K}\epsilon^2\right)) \times \left(1 - \frac{\delta(K-1)}{\gamma}\right)$$

**Remarks** Essentially, this result proves that the sample estimate of pairwise win ratios for previous running winners is a conservative estimate for the current running winner with high probability.

**Lemma 12 ($\epsilon$-optimality of running winner in *DE* w.r.t. eliminated items)** *An item $i$ is considered pairwise $\epsilon$-optimal w.r.t. an item $j$ iff. $p_{ij} > \frac{1}{2} - \epsilon$. Then, at any time step $t > 0$, $\forall j \in R(t)$, $i^*(t)$ is pairwise $\epsilon$-optimal w.r.t. $j$ with probability $1 - \frac{K\delta}{\gamma}$ where $K$ denotes the number of running winners $i^*(t)$ has inherited $(i^*(t), j)$ pairwise interactions from.*

**Proof**  We consider the different cases in which an item $j \in R(t)$ is eliminated.

- Case 1 - $N_{i^*(t)j} \geq m$, $P_{i^*(t)j} \geq \frac{1}{2} - \frac{\epsilon}{2}$: Applying Lemma 8 and Lemma 11, we have

$$Pr\left(\left(\frac{1}{2} - \frac{\epsilon}{2} - p_{i^*(t)j}\right) \geq \frac{\epsilon}{2}\right) \leq Pr\left((P_{i^*(t)j}(t) - p_{i^*(t)j}) \geq \frac{\epsilon}{2}\right)$$

$$\leq \frac{\delta}{\gamma} + \frac{(K-1)\delta}{\gamma} = \frac{K\delta}{\gamma}$$

$$\Rightarrow Pr\left(p_{i^*(t)j} \leq \frac{1}{2} - \epsilon\right) \leq \frac{K\delta}{\gamma}$$

$$\Rightarrow Pr\left(p_{i^*(t)j} > \frac{1}{2} - \epsilon\right) > 1 - \frac{K\delta}{\gamma}$$

- Case 2 - $U_{ji^*(t)} < 1/2$: We have $1/2 > U_{ji^*(t)} = P_{ji^*(t)} + \sqrt{\frac{\ln(\gamma/\delta)}{2N_{ji^*}}}$. It follows that $P_{i^*(t)j} = 1 - P_{ji^*(t)} \geq \frac{1}{2} + \sqrt{\frac{\ln(\gamma/\delta)}{2N_{ji^*(t)}}}$. Applying Lemma 8 and Lemma 11, we have for sample size $N \geq N_{ji^*(t)}$

$$\Rightarrow Pr\left(\left(\frac{1}{2} + \sqrt{\frac{\ln(\gamma/\delta)}{2N_{ji^*(t)}}} - p_{i^*(t)j}\right) \geq \sqrt{\frac{\ln(\gamma/\delta)}{2N_{ji^*(t)}}}\right)$$

$$\leq Pr\left((P_{i^*(t)j}(t) - p_{i^*(t)j}) \geq \sqrt{\frac{\ln(\gamma/\delta)}{2N_{ji^*(t)}}}\right)$$

$$\leq \frac{\delta}{\gamma} + \frac{(K-1)\delta}{\gamma} = \frac{K\delta}{\gamma}$$

$$\Rightarrow Pr\left(p_{i^*(t)j} \leq \frac{1}{2}\right) \leq \frac{K\delta}{\gamma}$$

$$\Rightarrow Pr\left(p_{i^*(t)j} > \frac{1}{2}\right) > 1 - \frac{K\delta}{\gamma}$$

$$\Rightarrow Pr\left(p_{i^*(t)j} > \frac{1}{2} - \epsilon\right) > 1 - \frac{K\delta}{\gamma}$$

∎

### E.4.2  PROOF OF SAMPLE COMPLEXITY UPPER BOUND

**Lemma 13 (Item elimination frequency)**  *Given some played set $G(t)$ of size $n_s$, it must be true that*

$$|Q(t + 2n_s(m-1)) \cap G(t)| \geq n_s - 1$$

*i.e., at least all but one item from the set will be eliminated in the next $n_s(m-1) + 2$ time steps.*

**Proof**  Let us first consider the following cases:

- Case 1 - $\forall j \in G(t) : N_{i^*(t)j} = 0$ (*i.e. running winner has not yet received pairwise updates with other items in the set*): In the next $n_s(m-1) + 1$ time steps, it must be true that at least one item in the set will have won at least $m$ times and $N_{ij} \geq m$ for all remaining items $j$ from $G(t)$. Let us denote this item $i$. Let us consider the following sub-cases:
  - If $i = i^*(t)$, all items that have not been eliminated earlier will be eliminated since $P_{i^*j} > 1/2$.
  - If $i \neq i^*(t)$, $i^*$ will be replaced, and only $i^*(t)$ will be removed. However, in the subsequent time step, since $N_{ij} \geq m$ for the remaining items $j$ from the original set $G(t)$ and $P_{ij} \geq 1/2$, these items will be eliminated.

  Consequently, all remaining items will be removed within $n_s(m-1) + 2$ time steps.

- Case 2 - $\exists j \in G(t): N_{i^*(t)j} \neq 0$ (*i.e. running winner has received pairwise updates for at least one other item in the set*): Let us again denote the set winner as $i$ and consider the following sub-cases:
    - If $i = i^*(t)$, then this case can be viewed as an intermediate stage of Case 1 and thus all 4 items will be removed in less than $n_s(m-1) + 2$.
    - If $i \neq i^*(t), N_{ii^*(t)}(t) = 0$, i.e. $i$ has not yet received pairwise updates with running winner at time step $t$, then in less than $n_s(m-1) + 1$ time steps, it will win $m$ times and all other items in the set will be eliminated since $N_{ij} \geq m, P_{ij} \geq 1/2$ for all remaining items $j$ from $G(t)$.
    - If $i \neq i^*(t), N_{ii^*(t)}(t) \neq 0$, then in less than $n_s(m-1) + 1$ time steps, it will win $m - N_{ii^*(t)}(t)$ more times and win the set, replacing $i^*(t)$ as the running winner. Let the time step this happens be denoted by $t'$. For items $j \in G(t): N_{ji^*(t)}(t) < N_{ii^*(t)}(t)$, if they have not been eliminated earlier, at $t'$, we will have $N_{ji^*(t)}(t') < m$ and thus this items will not be eliminated. In place of the eliminated item $i^*(t)$, a new item which we denote by $j'$ will be added. However, the new running winner $i^*(t')$ will inherit the pairwise interactions of the $i^*(t)$. Consequently, since $\sum_j N_{ji^*(t')} = \sum_j N_{ji^*(t)} > t' - t$, and as explained in Case 1, all items will be eliminated except the set winner before $\sum_j N_{ji^*(t')}$ reaches $k(m-1) + 1$, then, all items will be eliminated in $n_s(m-1) + 1$ time steps from $t$.

Consequently, the proof is complete. ∎

From Lemma 13, we can calculate the maximum number of time steps/iterations as $T = \lceil \frac{n}{n_s} \rceil \times (n_s(m-1) + 2)$. Given that for any replacement $i^*_{new}$ for the running winner $i^*$, we must have $N_{i^*_{new}i^*} \geq m$, the maximum number of unique running winners across all time steps is given by $\frac{T}{n_s(m-1)+2} = \lceil \frac{n}{n_s} \rceil$.

From Lemma 9, we can show by taking the intersection of all the probabilities that for any $0 \leq t, t' \leq T, t' > t, i^*(t') \neq i^*(t)$,

$$Pr\left(p_{i^*(t')i^*(t)} > \frac{1}{2}\right) > 1 - \frac{\delta}{\gamma} \times \left(\left\lceil \frac{n}{n_s} \right\rceil - 1\right) \tag{14}$$

since $\lceil \frac{n}{n_s} \rceil$ is the maximum number of running winners. Additionally, if we denote $i^*_\kappa$ as the $\kappa^{th}$ running winner, since the maximum of subsequent running winner changes is $\lceil \frac{n}{n_s} \rceil - \kappa$, then

$$Pr\left(p_{i^*(t')i^*_\kappa} > \frac{1}{2}\right) > 1 - \frac{\delta}{\gamma} \times \left(\left\lceil \frac{n}{n_s} \right\rceil - \kappa\right) \tag{15}$$

**Lemma 14 ($\epsilon$-optimality of $i^*$)** *In a finite number of time steps, the DE algorithm stops and returns an item $i^*$ such that*

$$Pr\left(p_{i^*j} > \frac{1}{2} - \epsilon\right) > 1 - \delta \tag{16}$$

**Proof** We note that there exists $t^* \leq T$ such that $i^*(t) = i^*$ for all $t \geq t^*$, i.e. the algorithm will return an $\epsilon$-optimal item within $T$ time steps. For any item $j \in S \setminus \{i^*\}$, there exists $t_j \leq t^*$ such that $j \in R(t_j)$. Applying Lemma 12 and using the transitivity property of the PL model (for all $i, j, k \in [n]$, if $p_{ij}, p_{jk} \geq \frac{1}{2}$, then $p_{ik} \geq \frac{1}{2}$ must be true as well), we have:

$$Pr\left(p_{i^*j} > \frac{1}{2} - \epsilon\right) \geq Pr\left(p_{i^*i^*(t_j)} > \frac{1}{2}\right) \times Pr\left(p_{i^*(t_j)j} > \frac{1}{2} - \epsilon\right)$$

$$\stackrel{(a)}{>} 1 - \frac{K\delta}{\gamma} \times \left(\left\lceil \frac{n}{n_s} \right\rceil - K\right) - \frac{\delta}{\gamma}$$

$$= 1 - \frac{\delta}{\gamma} \times \left( \left\lceil \frac{n}{n_s} \right\rceil \right)$$

$$\stackrel{(b)}{=} 1 - \delta$$

where inequality (b) holds true because $\gamma = \left\lceil \frac{n}{n_s} \right\rceil$. Hence Lemma 14 is proven. We note that $i^*(t_j)$ must be at least the $\kappa^{th}$ running winner and apply Eqn 15 for inequality (a). ∎

Lemma 14 states the $\epsilon$-optimality of the algorithm winner since it is pairwise $\epsilon$-optimal w.r.t. all items in $S$ including the true Condorcet winner. We now compute the sample complexity. This is straightforward since we have shown that the maximum number of time steps is

$$T = \lceil \frac{n}{n_s} \rceil \times (n_s(m-1) + 2)$$

$$\leq \left( (n + n_s)(m-1) + \frac{2(n + n_s)}{n_s} \right)$$

$$\leq \left( (n + n_s) \left( \frac{2 \ln((n/n_s + 1)/\delta)}{\epsilon^2} - 1 \right) + \frac{2(n + n_s)}{n_s} \right)$$

$$= \left( 2 \left( \frac{n + n_s}{\epsilon^2} \ln \left( \frac{n + n_s}{n_s \delta} \right) \right) + \frac{2(n + n_s)}{n_s} \right)$$

Consequently, the sample complexity is given by $O(\frac{n}{\epsilon^2} \ln(\frac{n}{n_s \delta}))$. We thus complete the proof of Theorem 1. ∎

### E.5 PROOFS OF ADDITIONAL SAMPLE COMPLEXITY RESULTS FOR *DE*

#### E.5.1 PROOF OF LEMMA 1

**Lemma 1 (Sample complexity lower bounds for DE)** *DE is $(\epsilon, \delta)$-PAC with best-case sample complexity* $O \left( \frac{n}{n_s} \ln \left( \frac{n}{n_s \delta} \right) \right)$.

**Proof** The correctness of *DE* will be proven in Appendix E.4.1. The best-case sample complexity corresponds to the case in which the final winner $i^*$ is selected in the initial item subset and it always wins the set. Under such an assumption, since an item $j \in [n] \backslash \{i^*\}$ will be eliminated when $U_{ji^*} < 1/2$ (Alg. 2: 2). Consequently, the number of timesteps required for elimination of the item $t_{\text{elim}}$ can be computed as follows:

$$U_{ji^*} = 0 + \sqrt{\frac{\ln(\gamma/\delta)}{2N_{ji^*}}} < \frac{1}{2}$$

$$\Rightarrow t_{\text{elim}} = \lceil 2 \ln(\gamma/\delta) \rceil$$

The maximum number of timesteps T can then be calculated as

$$T = \left\lceil \frac{n}{n_s} \right\rceil \times \left\lceil 2 \ln \left( \frac{\gamma}{\delta} \right) \right\rceil \leq \left( \frac{n}{n_s} + \frac{1}{2} \right) \times \left( 2 \ln \left( \frac{\gamma}{\delta} \right) + 1/2 \right)$$

The sample complexity is thus given by $O \left( \frac{n}{n_s} \ln \left( \frac{n}{n_s \delta} \right) \right)$.

#### E.5.2 PROOF OF LEMMA 2

The expected sample complexity for the *DE* algorithm is not well-defined since it is dependent on the reward distribution. For example, if the variance of the latent score distribution is very low, i.e. $\text{Var}(\theta_i) \sim 0$, for any two randomly sampled items $i$ and $j$, the win rate $p_{ij}$ is likely to be close to 1/2, i.e. $p_{ij}$ 1/2. In view of this, we compute a reward distribution dependent expected sample complexity

where the reward distribution is characterized by $\mathrm{Var}(p)$ which denotes the variance of $p_{ij}$ for any two randomly sampled items $i$ and $j$, i.e.

$$\mathrm{Var}(p) = \mathbb{E}\left[\left(p_{ij} - \frac{1}{2}\right)^2 \mid i, j \in [n]\right]$$

**Lemma 2 (Expected sample complexity for DE)** *Given a reward distribution such that $\mathrm{Var}(p) = V$, DE is $(\epsilon, \delta)$-PAC with an expected sample complexity upper bound of $O\left(\frac{n(1-V)}{\epsilon^2} \ln\left(\frac{n}{n_s \delta}\right)\right)$.*

**Proof** The correctness of *DE* will be proven in Appendix E.4.1. Given some item $i$ with win ratio respective to the running winner $p_{ii^*}$, assuming that only either $i$ and $i^*$ are winning, we can compute the timesteps required for item elimination $t_{\mathrm{elim}}^{ii^*}$ as follows:

$$U_{ii^*} = p_{ii^*} + \sqrt{\frac{\ln(\gamma/\delta)}{2N_{ii^*}}} < \frac{1}{2}$$

$$\Rightarrow t_{\mathrm{elim}}^{ii^*} = \left\lceil \frac{\ln(\gamma/\delta)}{2(1/2 - p_{ii^*})^2} \right\rceil$$

To obtain the actual $t_{\mathrm{elim}}$, we consider that for a subset of size $n_s$, the winning probability of the running winner is at least $1/n_s$ which yields $t_{\mathrm{elim}} \geq t_{\mathrm{elim}}^{ii^*} \times n_s$. Then, we have

$$t_{\mathrm{elim}} = \max\left(\left\lceil \frac{n_s \ln(\gamma/\delta)}{2(1/2 - p_{ii^*})^2} \right\rceil, m\right)$$

where $m = \frac{2\ln(\gamma/\delta)}{\epsilon^2}$ is the maximum number of updates before the item is considered a potential running winner challenger and either eliminated or promoted (Alg. 2: 2, 8-13). It is intractable to calculate the mean elimination time $\mathbb{E}(t_{\mathrm{elim}})$. However, with the upper bound on $t_{\mathrm{elim}}$, we can consider the random variable $X = (1/2 - p_{ii^*})^2$ ($\mathrm{Var}(p) = \mathbb{E}(X)$), and then

$$\mathbb{E}(t_{\mathrm{elim}}) = \frac{\ln(\gamma/\delta)}{2}\mathbb{E}\left(\frac{1}{X'}\right)$$

where $X'$ is lower bounded by $\epsilon^2/4n_s$ due to the $m$ upper bound. Consequently, we can obtain the following result using Jensen's inequality since $\mathbb{E}(X) < \mathbb{E}(X')$ and $X'$ has an upper bound of $1/4$:

$$\frac{2}{\ln(\gamma/\delta)}\mathbb{E}(t_{\mathrm{elim}}) \leq \frac{1/4 + \epsilon^2/4n_s - \mathrm{Var}(p)}{1/4 \times \epsilon^2/4n_s} = \frac{4n_s + \epsilon^2 - 4\mathrm{Var}(p)n_s}{\epsilon^2}$$

Consequently, expected number of timesteps T is bounded from above as follows:

$$T = \left\lceil \frac{n}{n_s} \right\rceil \times \frac{\ln(\gamma/\delta)}{2} \times \frac{4n_s + \epsilon^2 - 4n_s\mathrm{Var}(p)}{\epsilon^2}$$

$$\leq \left(n + \frac{n_s}{2}\right) \times \frac{\ln(\gamma/\delta)}{2} \times \frac{4 + \epsilon^2/n_s - 4\mathrm{Var}(p)}{\epsilon^2}$$

The expected sample complexity upper bound is thus $O\left(\frac{n(1-\mathrm{Var}(p))}{\epsilon^2} \ln\left(\frac{n}{n_s \delta}\right)\right)$.

### E.6 Proof of Lemma 15

**Lemma 15 (Supremacy of the winning partition)** *Given that the item correlation follows a $(r, c, c')$ noisy R-Block-Rank model and denoting WLOG the partition containing the winning item as $B_1 \ni 1$, if the following conditions are met:*

$$\mathbf{q} \cdot \mathbf{v}_1 \leq 1 - \varepsilon \,, \quad (c - c')(1 - \varepsilon) - \sqrt{2\varepsilon - \varepsilon^2} \left( \sqrt{1 - c'^2} + \sqrt{1 - c^2} \right) > \xi$$

*then for any item $i \in B_1$ and any item $j \notin B_1$, $\theta_i > \exp(\xi) \times \theta_j$ must be true.*

**Remarks**    This result is needed for the proof of Theorem 4. It allows us to define certain bounds within which the $(\epsilon - \delta)$-PAC condition can be met even in the worst-case scenarios since (as we will show in Appendix E.7) correctness of updates with respect to the winning partition is sufficient to guarantee the correctness of the *DEBC* algorithm.

**Proof**    We first state the following lemmas regarding general vector identities that will be used for this proof.

**Lemma 16** *Given unit vectors $\mathbf{q}, \mathbf{a}, \mathbf{b}$, $\mathbf{a} \cdot \mathbf{b} \leq c$, $\mathbf{q} \cdot \mathbf{a} \geq 1 - \epsilon$,*

$$\mathbf{q} \cdot (\mathbf{a} - \mathbf{b}) \geq (1 - c)(1 - \epsilon) - \sqrt{(1 - c^2)(2\epsilon - \epsilon^2)}$$

**Proof**    :

$$
\begin{aligned}
\mathbf{q} \cdot (\mathbf{a} - \mathbf{b}) &\geq \mathbf{q} \cdot (\mathbf{a} - (w_\| \mathbf{a} + w_\perp \mathbf{a}_\perp)) \\
&= (1 - w_\|)(\mathbf{q} \cdot \mathbf{a}) - w_\perp \mathbf{q} \cdot \mathbf{a}^\perp \\
&\geq (1 - c)(1 - \epsilon) - \sqrt{1 - c^2}\sqrt{1 - (1 - \epsilon^2)} \\
&= (1 - c)(1 - \epsilon) - \sqrt{(1 - c^2)(2\epsilon - \epsilon^2)}
\end{aligned}
$$

where

$$w_\| = \mathbf{a} \cdot \mathbf{b}, \quad w_\perp = \sqrt{1 - w_\|^2}, \quad \mathbf{a}_\perp = \frac{\mathbf{b} - w_\| \mathbf{a}}{|\mathbf{b} - w_\| \mathbf{a}|}$$

∎

**Lemma 17** *Given unit vectors $\mathbf{q}, \mathbf{a}, \mathbf{b}$, $\mathbf{a} \cdot \mathbf{b} \geq c$, $\mathbf{q} \cdot \mathbf{a} \geq 1 - \epsilon$,*

$$\mathbf{q} \cdot (\mathbf{a} - \mathbf{b}) \geq c(1 - \epsilon) - \sqrt{(1 - c^2)}$$

**Proof**

$$
\begin{aligned}
\mathbf{q} \cdot (\mathbf{a} - \mathbf{b}) &= \mathbf{q} \cdot (\mathbf{a} - (w_\| \mathbf{a} + w_\perp \mathbf{a}_\perp)) \\
&= (1 - w_\|)\mathbf{q} \cdot \mathbf{a} - w_\perp \mathbf{q} \cdot \mathbf{a}^\perp \\
&\leq (1 - c)(1 - \epsilon) + \sqrt{1 - c^2}\sqrt{1 - (1 - \epsilon^2)} \\
&= (1 - c)(1 - \epsilon) + \sqrt{(1 - c^2)(2\epsilon - \epsilon^2)}
\end{aligned}
$$

where

$$w_\| = \mathbf{a} \cdot \mathbf{b}, \quad w_\perp = \sqrt{1 - w_\|^2}, \quad \mathbf{a}_\perp = \frac{\mathbf{b} - w_\| \mathbf{a}}{|\mathbf{b} - w_\| \mathbf{a}|}$$

∎

**Lemma 18** *Given unit vectors $\mathbf{q}, \mathbf{x}, \mathbf{y}, \mathbf{z}$ such that:*

$$\mathbf{q} \cdot \mathbf{x} \geq 1 - \epsilon$$

$$\mathbf{x} \cdot \mathbf{y} \geq c$$
$$\mathbf{x} \cdot \mathbf{z} \leq c'$$

*Then, the following must be true:*

$$\mathbf{q} \cdot (\mathbf{y} - \mathbf{z}) \geq (c - c')(1 - \epsilon) - \sqrt{2\epsilon - \epsilon^2} \left( \sqrt{1 - c'^2} + \sqrt{1 - c^2} \right)$$

**Proof** Applying Lemma 16 and 17

$$\begin{aligned}
\mathbf{q} \cdot (\mathbf{y} - \mathbf{z}) &= \mathbf{q} \cdot (\mathbf{x} - \mathbf{z}) - \mathbf{q} \cdot (\mathbf{x} - \mathbf{y}) \\
&\geq \left( (1 - c')(1 - \epsilon) - \sqrt{(1 - c'^2)(2\epsilon - \epsilon^2)} \right) \\
&\quad - \left( (1 - c)(1 - \epsilon) + \sqrt{(1 - c^2)(2\epsilon - \epsilon^2)} \right) \\
&= (c - c')(1 - \epsilon) - \sqrt{2\epsilon - \epsilon^2} \left( \sqrt{1 - c'^2} + \sqrt{1 - c^2} \right)
\end{aligned}$$

∎

We can then apply Lemma 18 to the conditions in Lemma 15 which gives $\mathbf{q} \cdot (\mathbf{v}_i - \mathbf{v}_j) > \xi \Rightarrow \ln \theta_i > \ln \theta_j + \xi \Rightarrow \theta_i > \exp(\xi) \times \theta_j$ for any items $i \in B_1, j \notin B_1$.
∎

### E.7 PROOF OF THEOREM 4

**Theorem 4 (Sample complexity and correctness of *DEBC* with $R$-Block-Rank correlation)**
*Given that the item correlation follows a $R$-Block-Rank model and that the partition containing the winning item $B_1$ contains $n^*$ items, i.e. $|B_1| = n^*$, DEBC is $(\epsilon, \delta)$-PAC with worst-case sample complexity*

$$O \left( \max \left( \frac{\max(R, n_s \ln(n_s))}{w_{\min}^{in} \epsilon^2} \ln(\frac{n}{n_s \delta}) \; , \; \frac{n^*}{\epsilon^2} \ln(\frac{n^*}{n_s \delta}) \right) \right) \tag{3}$$

*given that the following conditions are met:*

1. $\mathbf{q} \cdot \mathbf{v}_1 \leq 1 - \varepsilon$

2. $(c - c')(1 - \varepsilon) - \sqrt{2\varepsilon - \varepsilon^2} \left( \sqrt{1 - c'^2} + \sqrt{1 - c^2} \right) > \ln \left( \frac{1 + 2\epsilon}{1 - 2\epsilon} \right)$

3. $1 - \frac{\delta n^*}{n + n_s} - \delta^{n_s - 1} > 1 - \delta$

4. $n^* + n_s \leq \left( \text{Info} \left( 1 - \frac{1}{\pi} \cos^{-1} \left( \frac{2 - 2c}{2(1 - c) + \lambda} \right) \right) \right)^{-1}$

**Interpretation of the conditions** Conditions 1 and 2 sets a lower bound on the score of the winning item as a function of the in-partition and cross-partition item correlations; it excludes the case in which all items are poorly correlated with the query which would limit the significance of the partitions. Condition 3 sets a bound on the size of the winning partition in relation to $n_s$ and $n$ in order for the probability bounds to be met, e.g. it excludes the case where $n^* \approx n$, i.e. almost all items fall into the same partition. Condition 4 places constraints on $n^*$ an $\lambda$ to avoid elimination of the wrong items from inferred updates in the worst case. We note that the results in Section B.2 show that this happens with very low probability. However, since we cannot obtain closed form solutions for this, Condition 4 is required.

**Remarks on the worst-case sample complexity** Assuming that $1/w_{\min}^{in}$ is small compared to the other factors, the sample complexity in this situation replaces the factor of $n$ in the general case with a factor of $n^*$, $R$ or $k \ln(n_s)$. Depending on the parameters of the $R$-Block-Rank model, this should be a large improvement. While the conditions may seem prohibitive, these are only required to create a structured item correlation through which lower bounds on the sample complexity can be proved.

### E.7.1 PROOFS FOR INTERMEDIATE RESULTS

**Proof** We first state the following extension to Theorem 2:

**Lemma 19** *Given any 3 partitions $B_\alpha, B_\beta, B_\omega$ and items $i, j \in B_\alpha, k \in B_\beta, h \in B_\omega$, the inferred update conditional probabilities are bounded as follows:*

$$p_{jk|ik}, p_{kj|ki} \geq 1 - \frac{1}{\pi} \cos^{-1}\left(\frac{c - 2c' + 1}{2(1 - c') + \lambda}\right)$$

$$p_{jk|hk}, p_{kj|kh} < 1 - \frac{1}{\pi} \cos^{-1}\left(\frac{c' + 1}{2(1 - c') + \lambda}\right)$$

**Proof** The first result can be obtained directly from Theorem 2. For the second result, we note that negative values in the item correlation matrix $C$ are set to zero in *DEBC* and apply Theorem 2 accordingly. ∎

Denoting for brevity $w^{in}_{\min}$ as

$$w^{in}_{\min} = \text{Info}\left(1 - \frac{1}{\pi} \cos^{-1}\left(\frac{c - 2c' + 1}{2(1 - c') + \lambda}\right)\right)$$

we can use Lemma 19 to prove the following results on partition elimination:

**Lemma 20 (Partition elimination by single winner)** *For any partition $B_\alpha$, if there exists item $i \notin B_\alpha$ that wins at least $\frac{2\ln(\gamma/\delta)}{\epsilon^2} \div w^{in}_{\min}$ sets containing any item from $B_\alpha$, then $B_\alpha$ will be entirely eliminated.*

**Proof** From Lemma 17, we have that the minimum conditional probability for intra-partition inferred updates is given by $1 - \frac{1}{\pi} \cos^{-1}\left(\frac{c-2c'+1}{2(1-c')}\right)$. Then, for any item $j \in B_\alpha$, we have that $N_{ij} \geq n_{i|\{i,j\}} \times w^{in}_{\min}$ according to the update step for **N** in Algorithm 3, where the lower bound corresponds to an item that has only received empirical updates and has not been played in a set. Since $i$ has not been eliminated despite having won more than $\frac{2\ln(\gamma/\delta)}{\epsilon^2}$ times, it is the running winner and hence $P_{ij} \geq (\frac{1}{2} - \frac{\epsilon}{2})$ if $N_{ij} \geq m \Rightarrow n_{i|\{i,j\}} \geq m \div w^{in}_{\min}$. Consequently, $j$ will be eliminated as an item that the running winner $i^*$ is at least pairwise $\epsilon$-optimal with. ∎

**Lemma 21 (Partition elimination from multiple winners)** *Let us denote $m'$ as*

$$m' = \frac{2\ln(\gamma/\delta)}{\epsilon^2} \div w^{in}_{\min}$$

*Then, for any partition $B_\alpha$, if there exists item $i \in B_\alpha$ that loses $(n_s - 1)(m' - 1) + 1$ sets won by any item not from $B_\alpha$, then either $B_\alpha$ will be entirely eliminated.*

**Proof** Across $(n_s - 1)(m' - 1) + 1$ losses, since there are $n_s - 1$ items in the set excluding the losing item, the running winner across the sets must have won at least $m'$ of those sets. Since the running winner inherits the pairwise interactions of the previous running winners, after $(n_s - 1)(m' - 1) + 1$ losses, denoting the running winner at that time step as $i^*$, all items from $B_\alpha$ have received at least $m'$ inferred or empirical updates with respect to $i^*$. By Lemma 20, $B_\alpha$ will be entirely eliminated. ∎

### E.7.2 PROOF OF SAMPLE COMPLEXITY UPPER BOUND

We can then proceed to analyze the sample complexity of *DEBC*. The algorithm will progress through two stages:

**Stage 1** Stage 1 is defined by the iterations during which multiple partitions still exist. From Lemma 21, a partition can accumulate a maximum of $(n_s - 1)(m' - 1) + 1$ losses to items from other sets before it is eliminated. Let us denote for brevity $\varrho = (n_s - 1)(m' - 1) + 1$. We consider two sub-stages:

1. Stage 1-A - *More than $n_s$ partitions remain*: In this stage, the set is created from minimally correlated items which ensures that items in the set are from different partitions. At each time step, $n_s - 1$ items lose the set to an item from a different partition. Since the losses can be distributed across $R$ partitions, we have that across $R \times m'$ time steps, $R \times m' \times (n_s - 1)$ losses are recorded in total, which means that each partition must have at least $m' \times (n_s - 1) > \varrho$ losses. Consequently, in less than $R \times m'$ time steps, $R - n_s + 1$ partitions will be removed and Stage 1-A ends.

2. Stage 1-B - *Less than $n_s$ but at least 2 partitions remain*: At the beginning of Stage 1-B, only $n_s - 1$ partitions remain. Let us denote by $t_r$ the time step at which there are only $r$ remaining partitions. We can then obtain the following expression:

$$t_{n_s-1} \geq \frac{(R - n_s + 1)\varrho}{n_s - 1}$$

$$t_r \geq t_{r+1} + \frac{\varrho}{r}$$

$$R\varrho - 1 \geq t_{n_s-1} \times (n_s - 1) + \sum_{r=1}^{r=n_s-2} (t_r - t_{r+1}) \times r \geq (R-1)\varrho$$

It is obvious that the maximum run time for Stage 1-B $\max_{t_1, t_2 \ldots t_{n_s-1}} \left( \sum_{r=1}^{r=n_s-1} t_r \right)$ is achieved by minimizing the rate at which losses are accumulated since the upper bound for the total losses $R\varrho - 1$ is fixed. This corresponds to partitions being removed as soon as possible up to the $t_2$, after which the losses are evenly split between the last two partitions to maximize the total accumulated losses. This yields

$$\sum_{r=1}^{r=n_s-1} t_r \leq \frac{(R - n_s + 1)\varrho}{n_s - 1} + \sum_{r=1}^{r=n_s-2} \frac{\varrho}{r}$$

$$\stackrel{(a)}{\leq} \frac{(R - n_s + 1)\varrho}{n_s - 1} + \varrho(\ln(n_s - 2) + 1)$$

$$< (R - n_s + 1)(m') + n_s m'(\ln(n_s + 1))$$

$$= m'(R - n_s + 1 + n_s \ln(n_s))$$

$$= \frac{2\ln(\gamma/\delta)}{\epsilon^2} \div w_{\min}^{in} \times (R - n_s + 1 + n_s \ln(n_s))$$

where for inequality (a), we note that the second term is a harmonic series and use the well-known result $\sum_{r=1}^{r=n_s} \frac{1}{n_s} \leq \ln(n) + 1$.

Hence, the sample complexity for stage 1 is $O\left( \frac{\max(R, n_s \ln(n_s))}{w_{\min}^{in} \epsilon^2} \ln\left(\frac{n}{\delta}\right) \right)$. We will revisit the unresolved term $w_{\min}^{in}$ later on.

**Stage 2** Stage 2 begins when there is only a single partition left. At this stage, we make the assumption that the inferred updates are insignificant. We validate this assumption in Lemma 22. Consequently, we can apply Theorem 1 which gives this step a sample complexity of $O(\frac{n^*}{\epsilon^2} \ln(\frac{n^*}{n_s \delta}))$.

Combining stages 1 and 2, *DEBC* with $R$-Block-Rank item correlation has a worst-case sample complexity of

$$O\left( \max\left( \frac{\max(R, n_s \ln(n_s))}{w_{\min}^{in} \epsilon^2} \ln\left(\frac{n}{n_s \delta}\right), \frac{n^*}{\epsilon^2} \ln\left(\frac{n^*}{n_s \delta}\right) \right) \right) \tag{17}$$

E.7.3    PROOF OF CORRECTNESS

The correctness of stage 2 is given by Theorem 1 as long as the remaining partition is in fact the winning partition. We now attempt to prove that this will indeed be the case under certain constraints:

**Lemma 22 (Resilience of the winning partition)** *If for any item $i \in B_1$ and any item $j \notin B_1$, $\theta_i > \exp(\xi) \times \theta_j$ such that*

$$\xi \geq \ln\left(\frac{1 + 2\epsilon}{1 - 2\epsilon}\right) \tag{18}$$

*then, the winning partition will be the last remaining partition with probability at least $1 - \delta^{n_s - 1}$.*

**Proof**  In order for Lemma 21 to result in the elimination of the winning partition $B_1$, it needs to lose to the running winner $\varrho$ times across a maximum of $2\varrho$ set plays (since if it wins the majority of those plays it becomes the running winner). Since the running winners are not items from $B_1$, denoting the minimum probability (across all item-pairs) that an item from $B_1$ beats the running winner as $p_{B_1 B_{\geq 2}}$, we have

$$p_{B_1 B_{\geq 2}} = \min_{i \in B_1, j \notin B_1} \left(\frac{\theta_i}{\theta_i + \theta_j}\right) = \frac{\exp(\xi)}{1 + \exp(\xi)}$$

Again, we can model the outcomes of the $2\varrho$ set plays as a sequence of Bernoulli trials with probability of success lower bounded by $p_{B_1 B_{\geq 2}}$. Then, denoting by $P_{B_1 B_{\geq 2}}$ the win rate of the item from $B_1$ over the running winner, we can apply Hoeffding's Inequality again to obtain

$$Pr\left(P_{B_1 B_{\geq 2}} \leq \frac{1}{2}\right)$$
$$\overset{(a)}{\leq} Pr\left(P_{B_1 B_{\geq 2}} \leq \frac{\exp(\xi)}{1 + \exp(\xi)} - \epsilon\right)$$
$$= Pr(P_{B_1 B_{\geq 2}} - p_{B_1 B_{\geq 2}} \leq -\epsilon)$$
$$\leq \exp(-2\varrho\epsilon^2) \leq \delta^{n_s - 1}$$

where Eqn. 18 can be algebraically manipulated to show $\frac{\exp(\xi)}{1+\exp(\xi)} - \epsilon \geq \frac{1}{2}$ for inequality (a).  ■

**Lemma 23 (In-partition conditional probability lower bounds)**  *Given any 3 items $i, j, k$ from the same partition, the inferred update conditional probabilities are bounded as follows:*

$$p_{jk|ik}, p_{kj|ki} \geq 1 - \frac{1}{\pi}\cos^{-1}\left(\frac{2 - 2c}{2(1 - c) + \lambda}\right)$$

**Proof**  The expression follows directly from Theorem 2.  ■

For Stage 2, we can use Theorem 1 together with Lemma 14 to show that it returns an $\epsilon$-optimal item from the last remaining partition with probability $1 - \frac{\delta n^*}{n + n_s}$ provided inferred updates are insignificant. For this to be true, the maximum $N_{ij}$ arising from inferred updates must be less than $m$ (to prevent item elimination). Denoting for brevity $w_{\max} = \text{Info}\left(1 - \frac{1}{\pi}\cos^{-1}\left(\frac{2-2c}{2(1-c)+\lambda}\right)\right)$, this is given by the condition

$$T \times w_{\max} \leq \frac{2\ln(\gamma/\delta)}{\epsilon^2} \Rightarrow n^* + n_s \leq \frac{1}{w_{\max}}$$

Since any item from the winning partition is $\epsilon$-optimal w.r.t. items from other partitions, the $\epsilon$-optimal item from the winning partition is also $\epsilon$-optimal w.r.t. all items. Consequently, the algorithm returns an $\epsilon$-optimal winner with probability at least

$$\left(1 - \frac{\delta n^*}{n + n_s}\right) \times (1 - \delta^{n_s - 1}) \geq 1 - \frac{\delta n^*}{n + n_s} - \delta^{n_s - 1}$$

Hence, the algorithm is $(\epsilon, \delta)$-PAC provided that $1 - \frac{\delta n^*}{n+n_s} - \delta^{n_s - 1} > 1 - \delta$.

# F  EXTENDING THE $\epsilon$-OPTIMAL ITEM TO THE GENERALIZED CONDORCET WINNER

In this section, we aim to draw a relation between PAC-best item identification and Generalized Condorcet winner (GCW) identification under the assumption of a PL model. Let us first define the following:

**Definition 1** *Given a set of items $[n]$, and item $i \in [n]$ is said to be the $k$-subset $\epsilon$-optimal Generalized Condorcet winner if and only if for all $G \subseteq [n], |G| = k$*

$$Pr(i|G) > \max_{j \in G}(Pr(j|G)) - \epsilon$$

*where $Pr(i|G)$ denotes the probability that item $i$ wins the set $G$.*

We then state and prove the following theorem:

**Theorem 5** *Given a set of items $[n]$, if an item $i$ is an $\epsilon$-optimal item, then it must also be a $k$-subset $\epsilon^*$ winner where $\epsilon^*$ is given by*

$$\epsilon^* = \frac{-4\epsilon}{k + 2\epsilon k - 4\epsilon}$$

**Proof** For any item $j \in G$, we have

$$\frac{\theta_i}{\theta_i + \theta_j} > \frac{1}{2} - \epsilon \Rightarrow \theta_i > \theta_j \times \frac{1 - 2\epsilon}{1 + 2\epsilon} \Rightarrow \theta_j > \theta_i \times \frac{1 + 2\epsilon}{1 - 2\epsilon}$$

Consequently, for any subset $G \in [n]$ of size $|G| = k$, we have for any item $j \in G$,

$$
\begin{aligned}
Pr(i|G) &= \frac{\theta_i}{\sum_{j \in G} \theta_j} \\
&> \frac{\theta_i}{\theta_i + \theta_i(k-1) \times \frac{1+2\epsilon}{1-2\epsilon}} \\
&= \frac{1 - 2\epsilon}{(k-1)(1+2\epsilon) + 1 - 2\epsilon} \\
&= \frac{1 - 2\epsilon}{k + 2\epsilon k - 4\epsilon} \\
Pr(j|G) &\overset{(a)}{=} \frac{p_{ji}}{p_{ij}} \times Pr(i|G) \\
&\leq \left(\frac{1 + 2\epsilon}{1 - 2\epsilon}\right) Pr(i|G)
\end{aligned}
$$

where we use the IIA property for equality (a). We can then combine both results to get

$$
\begin{aligned}
Pr(i|G) - Pr(j|G) &\geq \left(1 - \frac{1 + 2\epsilon}{1 - 2\epsilon}\right) \times Pr(i|G) \\
&> \frac{-4\epsilon}{1 - 2\epsilon} \times \frac{1 - 2\epsilon}{k + 2\epsilon k - 4\epsilon} \\
&= \frac{-4\epsilon}{k + 2\epsilon k - 4\epsilon}
\end{aligned}
$$

∎

Consequently, since *DE* finds an $\epsilon$-optimal item, and by extension, also a $k$-subset $\epsilon^*$-optimal GCW with probability $1 - \delta$, we argue that it is logical to compare it to an algorithm that also returns a $k$-subset $\epsilon^*$ GCW with probability $1 - \delta$. We suggest that the *Dvoretzky–Kiefer–Wolfowitz Tournament* (*DKWT*) algorithm (Haddenhorst et al., 2021) is such an algorithm under a slight modification - we introduce an early termination condition in the DKW mode-identification subroutine once the number of set plays is larger than $\frac{2 \ln(2/\delta)}{\epsilon^2}$ and return the mode. This is justified by the following result:

**Lemma 24** *Given a set of items $G$ has been played for $m = \frac{2 \ln(2/\delta)}{\epsilon^2}$ times, then the winning item must be the $\epsilon$-optimal Generalized Condorcet winner of the set, i.e.*

$$Pr(i|G) > \max_{j \in G}(Pr(j|G)) - \epsilon$$

**Proof** Let us denote the empirical win rate for each item $j \in G$ across $m$ plays by $p_{jG} = \frac{m_j}{m}$ where $m_j$ is the number of times item $j$ is selected. Then from the Dvoretzky–Kiefer–Wolfowitz inequality (Dvoretzky et al., 1956), we have

$$Pr\left(|p_{jG} - Pr(j|G)| > \frac{\epsilon}{2}\right) \leq 2e^{-m\epsilon^2/2} \tag{19}$$

Denoting the set winner across the $m$ plays by $i$, we have for all $j \in G \setminus \{i\}$ that $p_{iG} \geq p_{jG}$. Then, we have that

$$|p_{jG} - Pr(j|G)|, |p_{iG} - Pr(i|G)| \leq \frac{\epsilon}{2} \Rightarrow p_{iG} \geq p_{jG} - \epsilon$$

We then substitute $\delta = 2e^{-m\epsilon^2/2} \Rightarrow m = \frac{2\ln(2/\delta)}{\epsilon^2}$. Consequently, we have that given $m \geq \frac{2\ln(2/\delta)}{\epsilon^2}$, the following is true:

$$Pr\left(p_{iG} \geq p_{jG} - \epsilon\right) \geq 1 - \delta$$

which proves Lemma 24. ∎

In the mode-identification subroutine, a successful result indicates with high probability that the true winning probability of the winning item is at least $\epsilon^*$ higher than that of any item in the set. Lemma 24 shows that when the hardness parameter exceeds a certain threshold, the returned item is the $\epsilon^*$-optimal GCW of the subset with high probability $(1 - \delta)$.

We note that this is insufficient to guarantee correctness of the modified *DKWT* algorithm for the $\epsilon$-optimal GCW objective due to the changing prevailing winner which would require that each set winner is the $(\epsilon^*/\lceil n/k \rceil)$-optimal GCW and a different replacement condition for the prevailing winner (as in TTB (Saha & Gopalan, 2019c) and DE) to account for the worst case in which the prevailing winner is replaced in every set. However, we avoid modifying *DKWT* too drastically and use $m = \frac{2\ln(2/\delta)}{\epsilon^2}$ as a stopping criterion which should yield a conservative estimate for the sample complexity of *DKWT* (i.e. lower than if additional modifications were made to ensure correctness in the worst case scenario).

# G EXPERIMENT DETAILS AND ADDITIONAL RESULTS

## G.1 BASELINES

### G.1.1 SELECTED BASELINES

***Trace-the-Best (TTB)*** **and** ***Divide-and-Battle (DAB)*** Both of these algorithms were proposed in (Saha & Gopalan, 2019c) for $(\epsilon, \delta)$-PAC best-item identification and thus directly applicable to our setting.

*TTB* is based on randomly selecting item sets and maintaining a prevailing winner. Each set is played for the required number of rounds to determine the set winner before all losing items are eliminated from contention and a new set is selected from the remaining items to play against the prevailing winner. The sample complexity is not instance-dependent and is $O(\frac{n}{\epsilon^2}) \ln\left(\frac{n}{\delta}\right)$.

Like *TTB*, *DAB* similarly plays each set for a required number of times and eliminates all items except the winner. However, the sets are formed in a hierarchical fashion. It pre-divides the item set into subsets and plays each to obtain the winner, before dividing the winners into subsets and playing them against each other. The process is repeated until only one winner remains. (Saha & Gopalan, 2019c) proved an instance independent $O(\frac{n}{\epsilon^2}) \ln\left(\frac{k}{\delta}\right)$ sample complexity which is superior to that of *TTB*. However, when the constants are included, *DAB* has a significantly worse sample complexity than *TTB*.

***Dvoretzky–Kiefer–Wolfowitz Tournament (DKWT)*** This algorithm was proposed in (Haddenhorst et al., 2021) for identification of the Generalized Condorcet winner with relative feedback from

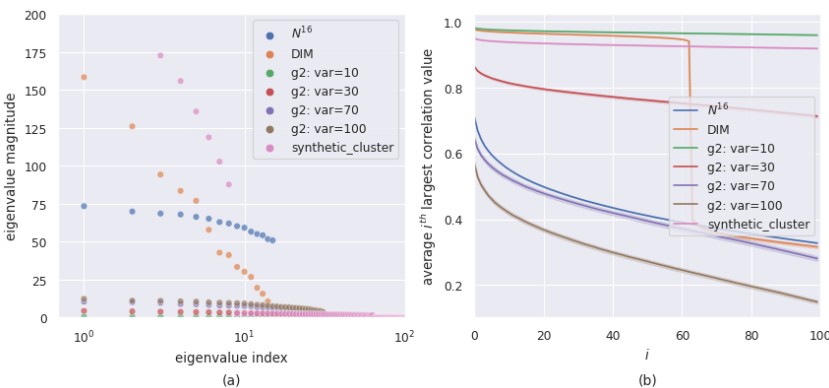

Figure 5: (a) Plot of eigenvalue magnitudes (sorted in descending order) (b) Plot of the mean of each item's $i^{th}$ largest correlation vector against $i$

fixed-sized subset plays in a general setting. Like *TTB*, it relies upon maintaining a prevailing winner and playing subsets to eliminate losing items in the set. However, it adaptively updates the hardness parameter to avoid excessive subset plays for simpler subsets where the winning item can be identified with fewer plays. To the best of our knowledge, this is the best existing baseline for best-item identification from fixed-sized subset plays that can be applied to the PL model. While it is not designed for the PAC setting, we show in Appendix F that an approximate equivalence can be established between the objectives of *DE* and *DKWT* under which we can compare the performance of the two algorithms.

### G.1.2  INCOMPATIBLE BASELINES

(Saha & Gopalan, 2020b) presents an instance optimal algorithm - *PAC wrapper* for obtaining the generalized Condorcet winner. However, (Haddenhorst et al., 2021) demonstrated that *DKWT* outperforms *PAC wrapper* by orders of magnitude in sample complexity and hence we include *DKWT* as a better baseline instead.

(Ren et al., 2021) present various algorithms for active ranking with multi-wise comparisons. However, while the work considers non deterministic feedback, it follows a fixed probability across all item subsets. More precisely, the comparisons are assumed to be correct with a certain probability $q > 2/3$. This is clearly incompatible with the PL model.

(Saha & Gopalan, 2019a) presents algorithms for full item ranking under winner or full subset ranking feedback with a PL model assumption, but this is incompatible with our objective of PAC best-item identification.

(Yang & Feng, 2023) presents an algorithm - *Nested Elimination* - for best-item identification from relative feedback from variable-sized subset plays. It assumes a general feedback model with the only requirement being that the item choice probabilities are consistent with some global item ranking. This is incompatible with our setting since there is no constraint on the subset size. In fact, the algorithm starts with playing all items in the set before gradually removing items from the played set.

### G.2  DATASETS

The correlation characteristics of each dataset are shown in Figure 5. Figure 5(a) plots the eigenvalue magnitudes in decreasing order for all used datasets while Figure 5(b) plots the mean (across all items) largest correlation values. For the $N^{16}$ dataset, we see that the 16 non-zero eigenvalues exhibit a gradual fall off which is consistent with the random initialization of the vectors. We also see that the highest correlation values are $< 0.8$. For the $d = 32$ DIM dataset, the correlation values show correlation values very close to 1 before a sharp fall off at $i = 63$ which corresponds to a cluster size of 64, i.e. each item is closely correlated to 63 other items. For the G2 datasets, we see lower correlation values for larger variance values. In particular, we see correlation values close to 1 for var=10 which indicates that all items in the same cluster are very closely correlated.

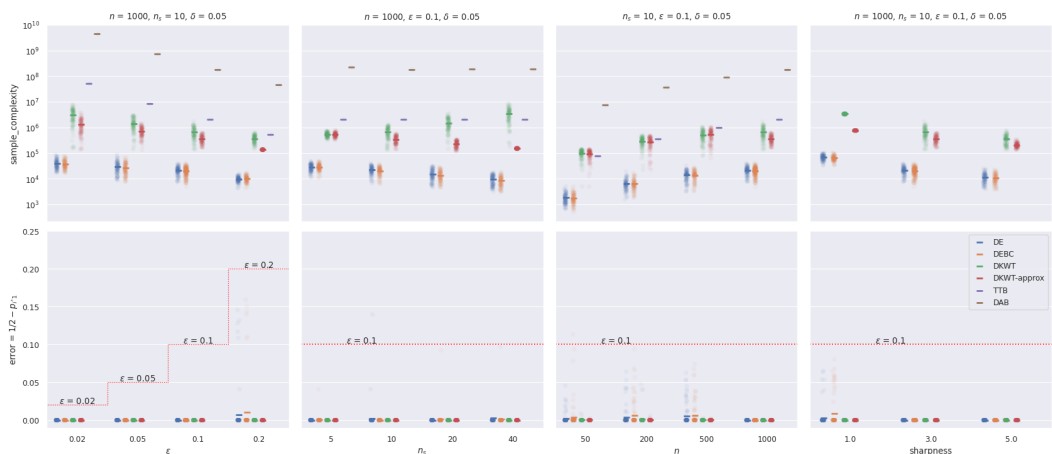

Figure 6: $N^{16}$ dataset: Sample complexity (first row) and error bias $\frac{1}{2} - p_{i*1}$ against $\epsilon$ across varying degrees of overlap

For each dataset, a common set of 100 query vectors are generated which are used to assess all algorithms where applicable. Each query vector is created by randomly selecting a vector from the dataset and perturbing it adding a random normal vector with norm = 0.4. This is to avoid the situation where the query vector is poorly correlated with the optimal item which is unlikely to be the ideal use case in practical applications (since a low score for all items indicates an indifference to the outcome).

### G.3 COMPUTE RESOURCES

Experiments were performed on an internal cluster with Intel® Xeon® E5-2698 v4 2.2 GHz CPUs. Evaluating the proposed algorithm for 100 trials required less than 5 hours for each setting. For the *DKWT* baseline, the evaluation was accelerated by the algorithm not having to make decisions at every time step which compensated for the higher sample complexity.

### G.4 ADDITIONAL RESULTS

Figures 6 and 7 show results from Section 8 but with their accompanying error biases, i.e. the degree of suboptimality of the algorithm winner given by $\frac{1}{2} - p_{i*1}$. The corresponding error bias hyperparameter $\epsilon$ is also plotted. Additionally, we also present the full set of experiments in Tables 1, 2 and 3. The mean values of sample complexity and error bias are given. The sample complexity standard deviation is given in brackets. The sucess rate refers to the proportion of trials for which the error bias is lower than $\epsilon$.

**Discussion on the validity of inferred updates in *DEBC*** While we see that *DE* and *DKWT* fulfil the $(\epsilon, \delta)$ PAC condition across all trials (in agreement with Theorem 1 which guarantees this for *DE*), *DEBC* fails to meet the $(1 - \delta)$ success rate in some experiments due to the probabilistic nature of the inferred updates. While preliminary analysis about the reliability of inferred updates can be found in Section 6, we leave more detailed analysis to future work. In particular, the results suggests that the reliability of inferred updates is dependent on the distribution of vectors in the datasets and their correlation characteristics. A more detailed study would ideally lead to methods to assign importance weights/thresholds to inferred updates in a dataset dependent manner. Nevertheless, we show that inferred updates in its current form can be directly used in scenarios where high accuracy is not the primary concern. In particular, we propose that inferred updates will be necessary in a sample-limited setting where the objective (ranking, best-item, etc.) has to be achieved with a limited number of samples.

**Discussion on *DE* sample complexity** We see that in many settings, the sample complexity of *DEBC* is only slightly better than *DE*. The exception to this is the DIM dataset for which *DEBC*

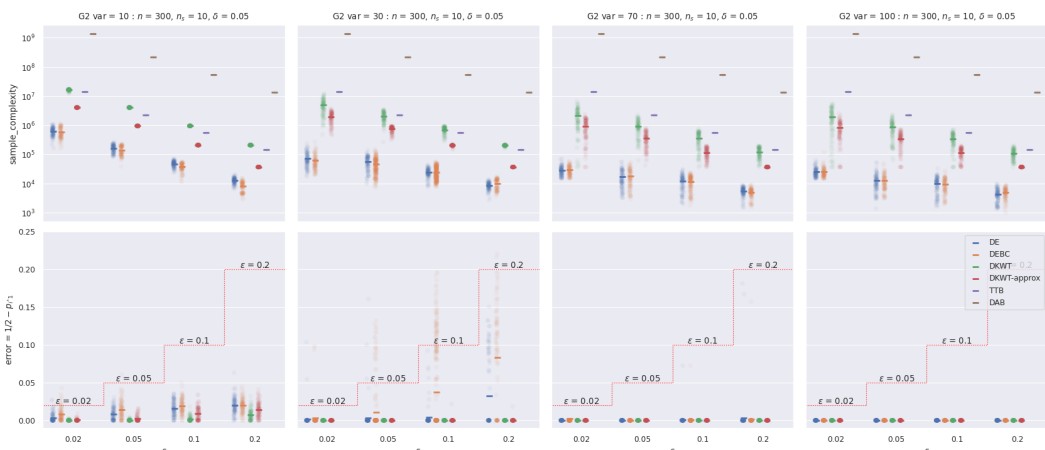

Figure 7: $d = 32$ G2 dataset: Sample complexity (first row) and error bias $\frac{1}{2} - p_{i*1}$ against $\epsilon$ across varying degrees of overlap

achieves significantly better sample complexity. Furthermore, we note that *DE* is vastly superior to *TTB* despite having a similar sample complexity upper bound (only superior by a $\ln k$) term. This suggests that the sample complexity upper bound in Theorem 1 might not be tight. At the very least, we postulate that a instance optimal sample complexity upper bound should exist. However, compared to other algorithms in which the static sets are evaluated with only the set winner persisting across sets, the fluid nature of *DE* poses significant challenges in deriving such a bound. We further postulate that a successful derivation of such an instance optimal sample complexity upper bound could also lead to a more general definition of the "hardness" of a dataset. We leave this as an important future work.

**Discussion on *DKWT* stopping criterion**     Setting the stopping criterion for $\epsilon$ according to the argument outlined in Appendix F yields very low error rates across all $\epsilon$ settings. While it is shown in Appendix F that the stopping criterion is set such that *DEBC* and *DKWT* are equivalent under the GCW identification objective, the excessively low error rates for *DKWT* indicates a sub-optimality for achieving this objective (i.e. it is unable to efficiently identify when to stop). To obtain a more competitive *DKWT* baseline, we introduce *DKWT-approx* as a baseline for which set the stopping criterion as $\epsilon$. We note that this baseline achieves the required error rates across all datasets, but emphasize that there is no guarantee for this. For example, a failure will occur in the worst case scenario where the set of items selected are all closely scored. In this scenario, an item that has selection probability within $\epsilon$ (guaranteed by the DKW inequality according to Eqn. 19) of that of the maximum item can still be less than $\epsilon^*$ optimal with respect to all the items.

Table 1: Complete experimental results for $N^{16}$ dataset

| $\epsilon$ | $n_s$ | n | $\delta$ | Algorithm | Sample Complexity | $\frac{1}{2} - p_{i*1}$ | success rate |
|---|---|---|---|---|---|---|---|
| 0.02 | 10 | 1000 | 0.05 | *DEBC* | 38090 (14269) | 0.000 | 1.000 |
| 0.02 | 10 | 1000 | 0.05 | *DE* | 39457 (15231) | 0.000 | 1.000 |
| 0.02 | 10 | 1000 | 0.05 | *DKWT* | 3122198 (1805390) | 0.000 | 1.000 |
| 0.05 | 10 | 1000 | 0.05 | *DEBC* | 26798 (13398) | 0.000 | 1.000 |
| 0.05 | 10 | 1000 | 0.05 | *DE* | 29986 (15160) | 0.000 | 1.000 |
| 0.05 | 10 | 1000 | 0.05 | *DKWT* | 1417514 (732469) | 0.000 | 1.000 |
| 0.10 | 5 | 1000 | 0.05 | *DEBC* | 28765 (8212) | 0.000 | 1.000 |
| 0.10 | 5 | 1000 | 0.05 | *DE* | 28084 (8394) | 0.000 | 1.000 |
| 0.10 | 5 | 1000 | 0.05 | *DKWT* | 527950 (100198) | 0.000 | 1.000 |
| 0.10 | 10 | 50 | 0.05 | *DEBC* | 1785 (843) | 0.004 | 0.990 |
| 0.10 | 10 | 50 | 0.05 | *DE* | 1827 (841) | 0.002 | 1.000 |
| 0.10 | 10 | 50 | 0.05 | *DKWT* | 96954 (31619) | 0.000 | 1.000 |
| 0.10 | 10 | 200 | 0.05 | *DEBC* | 6390 (3046) | 0.007 | 1.000 |
| 0.10 | 10 | 200 | 0.05 | *DE* | 6465 (3065) | 0.004 | 1.000 |
| 0.10 | 10 | 200 | 0.05 | *DKWT* | 288091 (105371) | 0.001 | 1.000 |
| 0.10 | 10 | 500 | 0.05 | *DEBC* | 13900 (5638) | 0.006 | 1.000 |
| 0.10 | 10 | 500 | 0.05 | *DE* | 13937 (5721) | 0.006 | 1.000 |
| 0.10 | 10 | 500 | 0.05 | *DKWT* | 513740 (230114) | 0.000 | 1.000 |
| 0.10 | 10 | 1000 | 0.05 | *DEBC* | 27164 (20794) | 0.002 | 1.000 |
| 0.10 | 10 | 1000 | 0.05 | *DE* | 30667 (24020) | 0.001 | 0.997 |
| 0.10 | 10 | 1000 | 0.05 | *DKWT* | 1254436 (1239332) | 0.000 | 1.000 |
| 0.10 | 20 | 1000 | 0.05 | *DEBC* | 13712 (5960) | 0.001 | 1.000 |
| 0.10 | 20 | 1000 | 0.05 | *DE* | 15234 (6938) | 0.000 | 1.000 |
| 0.10 | 20 | 1000 | 0.05 | *DKWT* | 1447721 (728557) | 0.000 | 1.000 |
| 0.10 | 40 | 1000 | 0.05 | *DEBC* | 8499 (3549) | 0.001 | 1.000 |
| 0.10 | 40 | 1000 | 0.05 | *DE* | 9425 (3829) | 0.003 | 0.990 |
| 0.10 | 40 | 1000 | 0.05 | *DKWT* | 3394139 (1910341) | 0.000 | 1.000 |
| 0.20 | 10 | 1000 | 0.05 | *DEBC* | 10088 (2708) | 0.010 | 0.990 |
| 0.20 | 10 | 1000 | 0.05 | *DE* | 9595 (2187) | 0.007 | 1.000 |
| 0.20 | 10 | 1000 | 0.05 | *DKWT* | 367515 (117279) | 0.000 | 1.000 |

Table 2: Complete experimental results for $d = 32$ DIM dataset

| $\epsilon$ | $n_s$ | n | $\delta$ | Algorithm | Sample Complexity | $\frac{1}{2} - p_{i*1}$ | success rate |
|---|---|---|---|---|---|---|---|
| 0.02 | 10 | 1024 | 0.05 | *DEBC* | 288951 (142524) | 0.010 | 0.828 |
| 0.02 | 10 | 1024 | 0.05 | *DE* | 493839 (198555) | 0.003 | 0.980 |
| 0.02 | 10 | 1024 | 0.05 | *DKWT* | 16282011 (3621992) | 0.000 | 1.000 |
| 0.05 | 10 | 1024 | 0.05 | *DEBC* | 124655 (34674) | 0.012 | 0.990 |
| 0.05 | 10 | 1024 | 0.05 | *DE* | 132618 (26189) | 0.010 | 1.000 |
| 0.05 | 10 | 1024 | 0.05 | *DKWT* | 6219200 (626776) | 0.000 | 1.000 |
| 0.10 | 10 | 1024 | 0.05 | *DEBC* | 23704 (9524) | 0.024 | 1.000 |
| 0.10 | 10 | 1024 | 0.05 | *DE* | 38997 (5786) | 0.014 | 1.000 |
| 0.10 | 10 | 1024 | 0.05 | *DKWT* | 1648601 (89310) | 0.001 | 1.000 |
| 0.20 | 10 | 1024 | 0.05 | *DEBC* | 6865 (4719) | 0.024 | 1.000 |
| 0.20 | 10 | 1024 | 0.05 | *DE* | 11306 (1653) | 0.020 | 1.000 |
| 0.20 | 10 | 1024 | 0.05 | *DKWT* | 423978 (18085) | 0.003 | 1.000 |

Table 3: Complete experimental results for $d = 32$ G2 dataset

| var | $\epsilon$ | $n_s$ | n | $\delta$ | Algorithm | Sample Complexity | $\frac{1}{2} - p_{i^*1}$ | success rate |
|---|---|---|---|---|---|---|---|---|
| 10 | 0.02 | 10 | 300 | 0.05 | DEBC | 573297 (238819) | 0.008 | 0.898 |
| 10 | 0.02 | 10 | 300 | 0.05 | DE | 619373 (194185) | 0.004 | 0.980 |
| 10 | 0.02 | 10 | 300 | 0.05 | DKWT | 16095899 (960872) | 0.000 | 1.000 |
| 10 | 0.05 | 10 | 300 | 0.05 | DEBC | 140637 (48557) | 0.014 | 0.990 |
| 10 | 0.05 | 10 | 300 | 0.05 | DE | 160539 (45630) | 0.008 | 1.000 |
| 10 | 0.05 | 10 | 300 | 0.05 | DKWT | 4018854 (57767) | 0.000 | 1.000 |
| 10 | 0.10 | 10 | 300 | 0.05 | DEBC | 37111 (10780) | 0.019 | 1.000 |
| 10 | 0.10 | 10 | 300 | 0.05 | DE | 47064 (10442) | 0.016 | 1.000 |
| 10 | 0.10 | 10 | 300 | 0.05 | DKWT | 940636 (7077) | 0.002 | 1.000 |
| 10 | 0.20 | 10 | 300 | 0.05 | DEBC | 8073 (3074) | 0.020 | 1.000 |
| 10 | 0.20 | 10 | 300 | 0.05 | DE | 12477 (2796) | 0.020 | 1.000 |
| 10 | 0.20 | 10 | 300 | 0.05 | DKWT | 205798 (1824) | 0.007 | 1.000 |
| 30 | 0.02 | 10 | 300 | 0.05 | DEBC | 61441 (39533) | 0.003 | 0.970 |
| 30 | 0.02 | 10 | 300 | 0.05 | DE | 73432 (40950) | 0.002 | 0.980 |
| 30 | 0.02 | 10 | 300 | 0.05 | DKWT | 4846755 (2611789) | 0.000 | 1.000 |
| 30 | 0.05 | 10 | 300 | 0.05 | DEBC | 45199 (27385) | 0.013 | 0.879 |
| 30 | 0.05 | 10 | 300 | 0.05 | DE | 55288 (26533) | 0.004 | 0.970 |
| 30 | 0.05 | 10 | 300 | 0.05 | DKWT | 1990971 (736651) | 0.000 | 1.000 |
| 30 | 0.10 | 10 | 300 | 0.05 | DEBC | 24834 (11949) | 0.040 | 0.793 |
| 30 | 0.10 | 10 | 300 | 0.05 | DE | 23816 (7679) | 0.004 | 1.000 |
| 30 | 0.10 | 10 | 300 | 0.05 | DKWT | 688586 (166016) | 0.000 | 1.000 |
| 30 | 0.20 | 10 | 300 | 0.05 | DEBC | 9966 (3782) | 0.083 | 0.979 |
| 30 | 0.20 | 10 | 300 | 0.05 | DE | 8378 (2210) | 0.032 | 1.000 |
| 30 | 0.20 | 10 | 300 | 0.05 | DKWT | 200149 (8018) | 0.000 | 1.000 |
| 70 | 0.02 | 10 | 300 | 0.05 | DEBC | 28707 (10207) | 0.003 | 0.990 |
| 70 | 0.02 | 10 | 300 | 0.05 | DE | 28271 (10464) | 0.000 | 1.000 |
| 70 | 0.02 | 10 | 300 | 0.05 | DKWT | 2111053 (1438009) | 0.000 | 1.000 |
| 70 | 0.05 | 10 | 300 | 0.05 | DEBC | 18223 (11575) | 0.000 | 1.000 |
| 70 | 0.05 | 10 | 300 | 0.05 | DE | 16690 (10883) | 0.000 | 1.000 |
| 70 | 0.05 | 10 | 300 | 0.05 | DKWT | 913191 (491337) | 0.000 | 1.000 |
| 70 | 0.10 | 10 | 300 | 0.05 | DEBC | 11615 (5526) | 0.001 | 1.000 |
| 70 | 0.10 | 10 | 300 | 0.05 | DE | 12020 (6458) | 0.001 | 1.000 |
| 70 | 0.10 | 10 | 300 | 0.05 | DKWT | 351023 (184430) | 0.000 | 1.000 |
| 70 | 0.20 | 10 | 300 | 0.05 | DEBC | 4901 (1664) | 0.002 | 1.000 |
| 70 | 0.20 | 10 | 300 | 0.05 | DE | 5422 (1630) | 0.003 | 1.000 |
| 70 | 0.20 | 10 | 300 | 0.05 | DKWT | 118016 (44772) | 0.000 | 1.000 |
| 100 | 0.02 | 10 | 300 | 0.05 | DEBC | 25321 (12164) | 0.000 | 1.000 |
| 100 | 0.02 | 10 | 300 | 0.05 | DE | 25316 (9172) | 0.000 | 1.000 |
| 100 | 0.02 | 10 | 300 | 0.05 | DKWT | 1937737 (1431587) | 0.000 | 1.000 |
| 100 | 0.05 | 10 | 300 | 0.05 | DEBC | 12502 (7960) | 0.000 | 1.000 |
| 100 | 0.05 | 10 | 300 | 0.05 | DE | 12590 (8568) | 0.000 | 1.000 |
| 100 | 0.05 | 10 | 300 | 0.05 | DKWT | 851744 (517000) | 0.000 | 1.000 |
| 100 | 0.10 | 10 | 300 | 0.05 | DEBC | 9257 (4973) | 0.000 | 1.000 |
| 100 | 0.10 | 10 | 300 | 0.05 | DE | 9989 (5647) | 0.000 | 1.000 |
| 100 | 0.10 | 10 | 300 | 0.05 | DKWT | 339929 (160494) | 0.000 | 1.000 |
| 100 | 0.20 | 10 | 300 | 0.05 | DEBC | 4866 (1784) | 0.000 | 1.000 |
| 100 | 0.20 | 10 | 300 | 0.05 | DE | 4284 (2037) | 0.000 | 1.000 |
| 100 | 0.20 | 10 | 300 | 0.05 | DKWT | 106218 (39186) | 0.000 | 1.000 |

