# OpenReview forum: "Dynamic Elimination For PAC Optimal Item Selection From Relative Feedback"
_ICLR.cc/2025/Conference — Submitted to ICLR 2025_

### Official Review · Reviewer_Dfab · 2024-11-04

**Soundness:** 3
**Presentation:** 3
**Contribution:** 2
**Rating:** 3
**Confidence:** 4

**Summary:**

This paper studies the best item selection problem from noisy multi-wise comparisons. There are totally n items, and at each round the agent can select n_s items to compare, and the comparison will return one item as the winner according to the PL model. The better item will have a higher chance to win the comparison. The problem is to find the best item with 1-\delta confidence with the least amount of comparisons.

The authors propose a new algorithm for best item selection from multi-wise comparisons and give the worst-case, best-case, and expected sample complexity. The authors further propose a method to estimate the winning probabilities between two items without directly comparing these two items.

**Strengths:**

This paper proposes new algorithms for best item identification and winning rate estimation. These algorithms do have some novelty and are inspiring.

**Weaknesses:**

The significance of this paper's results is questionable. For the best item identification, the sample complexity (worst-case) is O(n*\epsilon^{-2} *\log(n*n_s^{-1}*\delta^{-1})). However, in a previous paper [1], when n_s = 2 (i.e., pairwise comparisons), the sample complexity (expected) for best item identification is O(n*\epsilon^{-2} *\log\delta^{-1})) (Theorem 5 of [1]), which is log(n) better than the proposed algorithm. When n_s is large enough like \Omega(n), the sample complexity of the proposed in this paper will become the same as that in [1]. Hence, the proposed algorithm does not show superiority compared to existing algorithms, or at least be as good as existing ones. Although its sample complexity is worst case instead of in expectation, but this difference is not large enough to support the significance of the new results.

Besides, the winning chance estimation and the sample complexity of best item identification seem not to be correlated enough to be put in the same paper. Putting them in one paper makes the paper's scope to be ambiguous. If the winning chance estimate is significant enough, it is better to be placed in another paper focusing on a more related topic.

[1] Ren, W., Liu, J., & Shroff, N. (2020, November). The Sample Complexity of Best-$ k $ Items Selection from Pairwise Comparisons. In International Conference on Machine Learning (pp. 8051-8072). PMLR.

**Questions:**

It will help if the authors can have more evidence to demonstrate this paper's results' significance.
Or if I perceive this paper wrongly, please let me know.

---

> ### Author Response · Authors · 2024-11-15
> **Response to Reviewer Dfab**
>
> We would first like to thank Reviewer Dfab for his feedback and time taken to review the paper.
>
> We respond to the weaknesses raised by Reviewer Dfab as follows:
>
> > The significance of this paper's results is questionable. For the best item identification, the sample complexity (worst-case) is O(n*\epsilon^{-2} \log(nn_s^{-1}\delta^{-1})). However, in a previous paper [1], when n_s = 2 (i.e., pairwise comparisons), the sample complexity (expected) for best item identification is O(n\epsilon^{-2} *\log\delta^{-1})) (Theorem 5 of [1]), which is log(n) better than the proposed algorithm. ....
>
> This paper was mentioned in Appendix G.1.2. as an incompatible baseline because it considers a fixed probability across all item subsets, i.e. the subset plays are assumed to yield the correct results with a fixed probability p>2/3. In contrast, the PL model assumes a latent score distribution across all items. Consequently, if the optimal item has a similar score to another item in the set, the wrong item will be returned with higher probability which arguably makes the task more challenging.
>
> > Besides, the winning chance estimation and the sample complexity of best item identification seem not to be correlated enough to be put in the same paper. Putting them in one paper makes the paper's scope to be ambiguous. If the winning chance estimate is significant enough, it is better to be placed in another paper focusing on a more related topic.
>
> The algorithms in the paper yield an ε-optimal item with probability $1-\delta$, where an item is ε optimal if the probability that it beats the actual winning item 1 is larger than 1/2 − ε. There is no winning rate estimation.
>
> > It will help if the authors can have more evidence to demonstrate this paper's results' significance. Or if I perceive this paper wrongly, please let me know.
>
> We have demonstrated superior sample complexity to existing comparable SOTA algorithms and far superior sample complexity in practice by considering synthetic datasets of various distributions that we argue covers the main scenarios for vector distributions in line 410-416 (we discuss these in greater detail in Appendix G.2.).

---

> > ### Comment · Reviewer_Dfab · 2024-11-20
> > **Response to the rebuttal**
> >
> > Thanks for your response.
> >
> > For point 1, in [1], you can check Section 1.2 the formulation. The formulation does not require a fixed probability. The only requirement is pij > 1/2 if i is better than j. When n_s = 2, the PL model is a special case of the formulation considered in [1]. The proposed algorithms have at least a log gap between the bounds in [1].
> >
> > For point 2, I am still confused by we put sample complexity works and winning rate estimation in a same paper.

---

### Official Review · Reviewer_DQ7C · 2024-11-04

**Soundness:** 2
**Presentation:** 2
**Contribution:** 2
**Rating:** 3
**Confidence:** 3

**Summary:**

This paper studies the problem of best-item identification from relative feedback in the setting where a learner adaptively plays subsets of items and receives stochastic feedback in the form of the best item in the set. An algorithm named Dynamic Elimination (DE) is proposed, which dynamically prunes sub-optimal items from contention to efficiently identify the best item. Then the model is extended to capture the generalized linear correlation of items. An algorithm named DEBC, an extension of DE is proposed to handle this extension. The core idea is leveraging the generalized linear correlation to obtain estimates on item win rates without directly playing them by leveraging item correlation information. Extensive experiments are conducted to validate the empirical performance of the proposed algorithm.

**Strengths:**

This paper studies an important problem.

The proposed algorithm is complemented with theoretical analysis as well as extensive experiments.

**Weaknesses:**

The algorithmic contribution of this looks narrow. The core idea of DE is flexibly eliminating items once they are deemed suboptimal. This idea is old in the bandit literature. The authors can refer to Chapter 6 of [3] for some reference.  Also, a simple google search would gives you a number of work on elimination algorithms. The core idea of DEBC is exploiting the generalized linear structure on the correlation among arms. This idea is also not new since linear structure has been extensively studied in linear bandits, reinforcement learning. Please refer to Part V of [3] from some details.

The proof techniques of this paper are not new, most of them are drawn from literature. Thus, this paper does not contribute to new proof techniques.  To me more specific, compared to [1,2], I do not see enough new ideas in the proof.  For example, the analysis of concentration, probability of event, etc., looks very normal. Could the author elaborate on the novelty of the proof?

The theoretical improvement over SOTA techniques is not clear. The improvement on the sample complexity compared with SOTA works is not stated.  How does it improve the sample complexity upper bound?

The second paragraph of the related work overstated the limitations of previous works without any supporting evidence. Previous algorithms may require up to millions of samples to rank only a few items, but this possibility depends on the setting of the problem. It should not be stated as a general claim. Furthermore, this paragraph is not precise. What do you mean by often? Could you quantify it?

Lemma 1 is confusing. It is highlighted as a lower bound, but the sample complexity is stated using the big O notation.

[1] Yisong Yue, et al. The K-armed Dueling Bandits Problem, Journal of Computer and System Sciences, 78(5): 1538–1556.

[2] Björn Haddenhorst. Identification of the Generalized Condorcet Winner in Multi-dueling Bandits, NeurIPS, 2021

[3] Tor Lattimore, et al. Bandit Algorithms. Cambridge press.

**Questions:**

See weakness part.

---

> ### Author Response · Authors · 2024-11-15
> **Response to Reviewer DQ7C**
>
> We would first like to thank Reviewer DQ7C for his feedback and time taken to review the paper.
>
> We respond to the weaknesses raised by Reviewer DQ7C as follows:
>
> >The algorithmic contribution of this looks narrow. The core idea of DE is flexibly eliminating items once they are deemed suboptimal. This idea is old in the bandit literature ... The core idea of DEBC is exploiting the generalized linear structure on the correlation among arms. This idea is also not new ...
>
> We agree that early elimination of suboptimal items and exploiting a linear structure is generally speaking not new to bandits. However, we are exploring the problem of best-item identification from **relative feedback** by playing fixed subsets.
>
> Regarding early item elimination, all existing SOTA algorithms that investigate this issue directly (a detailed discussion about these algorithms and why certain baselines are not comparable is in Appendix G.1) play a fixed subset repeatedly until the set optimal item is found before eliminating all other items in the set. The reason for this is because elimination of items before the set optimal item has been found renders the pairwise win ratios irrelevant and the win counts need to be accumulated again for the other items in the set. This is precisely where our algorithm differs because we allow elimination of items in the set by introducing the novel idea of inheriting pairwise win statistics under certain conditions and proving its validity (running winner inheritance discussed in lines 224-229 and proved in Appendix E.4, Lemma 9-10). In practice, this can yield huge sample complexity improvements since challenging sets (where there are two or more items with very similar win ratios) need to be played a large number of times to differentiate the two competitive items; our algorithm allows other items to be eliminated and replaced while this is happening. This is reflected in the sample complexity improvements demonstrated in our experiments. We welcome further discussion on this point.
>
> Regarding exploiting a linear structure, we are again the first (to the best of our knowledge) to investigate this in the relative feedback from fixed-size subset plays setting. We present a novel of inferred updates by investigating the question: Given what we know about items i, j and k, if item i is ranked above/below item k, how likely is it that item j is ranked above/below item k? This has only been investigated in a few works which we have discussed in the related works section (line 93-101), but not in the setting of best-item identification from fixed-size subset plays. Moreover, we do not require knowledge of the item's precise vector representation but only the item correlations. We further discuss extensively the limitations of our results in Appendix B.
>
> > Could the author elaborate on the novelty of the proof?
> As discussed above, the algorithm is itself novel (dynamically eliminating items instead of the whole subset) and relies on the validity of running winner inheritance which is a new result. We prove these results in the process of proving correctness and obtaining sample complexity bounds on our algorithm. Insofar as the algorithm is fundamentally very different from the algorithms in [1] and [2], I am not unsure how the proof can be considered as similar although it does rely on similar techniques.
>
> > The theoretical improvement over SOTA techniques is not clear. The improvement on the sample complexity compared with SOTA works is not stated. How does it improve the sample complexity upper bound?
>
> We improve against the existing algorithms by a factor of ln(n/ns). This is mentioned in the Appendix but we will explicitly mention the comparison in the main section. However, in practice, the margin of improvement is much larger as shown in the experiments.
>
> > The second paragraph of the related work overstated the limitations of previous works without any supporting evidence. Previous algorithms may require up to millions of samples to rank only a few items, but this possibility depends on the setting of the problem. ...
>
> The two SOTA algorithms that are directly compatible with our setting are TTB and DAB and these are non instance optimal algorithms, hence we can determine their sample complexities beforehand. These are presented in Section 8 and are in the 10^7-10^10 range for a few hundred items. Comparing instance optimal algorithms, we used a modified version of DKWT, but the directly compatible PAC wrapper [4] is shown in [2] to require more than 10^8 samples to obtain the Generalised Condorcet Winner.
>
> > Lemma 1 is confusing. It is highlighted as a lower bound, but the sample complexity is stated using the big O notation.
>
> We apologise for the typo and will correct it.
>
> [4] Aadirupa Saha and Aditya Gopalan. From pac to instance-optimal sample complexity in the plackett- luce model. In International Conference on Machine Learning, pp. 8367–8376. PMLR, 2020b.

---

### Official Review · Reviewer_2tcm · 2024-11-04

**Soundness:** 2
**Presentation:** 2
**Contribution:** 2
**Rating:** 5
**Confidence:** 3

**Summary:**

The authors have introduced a dynamic elimination method for item selection based on relative feedback. They propose two distinct algorithms: one that implements the fundamental dynamic elimination approach and another that incorporates item correlations into the elimination process. The paper provides theoretical analysis on both the sample complexities and the correctness of the proposed methods. Furthermore, the authors demonstrate the proposed methods' ability in terms of reducing sample complexities when compared to several baseline approaches.

**Strengths:**

S1: The proposed methods enhance existing approaches by dynamically eliminating suboptimal items, significantly reducing the algorithm's complexity.

S2: By incorporating correlations between items, the proposed methods extend their applicability to items initially absent from the played set.

S3: The authors offer theoretical assurances regarding the sample complexity of DE and DEBC. They also demonstrate that the sample mean of an inferred update sequence serves as an unbiased estimator.

**Weaknesses:**

W1: The DE and DEBC algorithms are designed for the task of item selection, yet the performance of best item identification is neglected. The authors dedicate substantial space to discussing the efficiency of the proposed methods in reducing sample complexity. However, the mathematical formulation lacks clarity, as the best item identification problem and the relative feedback are not thoroughly formulated.
W2: The proposed DEBC algorithm presumes that item correlations are known to the user, raising concerns about the validity of this assumption. The authors do not address the implications of this assumption in real-world applications or indicate whether it is a common assumption in existing literature.
W3: The current work lacks demonstration in real-world scenarios. Although the authors mention that learning to rank is crucial in fields like sociology, information retrieval, and search engine optimization, they do not provide examples of its application in these areas. Consequently, the practical applicability of this work remains uncertain.

**Questions:**

Q1: It would be appreciated if the authors could include additional experiments conducted in real-world scenarios. These experiments should aim to demonstrate the effectiveness of the proposed methods in best item identification. Additionally, it is recommended that results be presented using widely accepted metrics such as accuracy, AUC, F1, precision, and recall.
Q2: The authors might consider providing a more precise and explicit formulation of the item selection problem in Section 3. Furthermore, it would be valuable to discuss in greater detail how DE and DEBC can be applied in practical, real-world contexts.

---

### Official Review · Reviewer_3V2W · 2024-11-05

**Soundness:** 3
**Presentation:** 2
**Contribution:** 2
**Rating:** 6
**Confidence:** 2

**Summary:**

This work addresses the problem of identifying the best item from a set of items based on relative feedback, specifically using a method called Dynamic Elimination (DE). DE efficiently prunes sub-optimal items as it progresses, improving sample complexity compared to existing algorithms. The authors also propose an extension, Dynamic Elimination by Correlation (DEBC), which incorporates inferred updates based on item correlations. DEBC significantly outperforms DE in settings where item correlation is strong, reducing sample complexity further. Extensive experiments demonstrate that both DE and DEBC outperform existing state-of-the-art (SOTA) methods in terms of sample complexity across multiple datasets and settings. Additionally, the paper explores future directions for improving sample complexity bounds and extending the methods to partial/full rankings.

**Strengths:**

1. DE and its extension, DEBC, significantly improve sample complexity for identifying the best item compared to existing algorithms, reducing the number of subset plays needed.

2. The incorporation of inferred updates through item correlation in DEBC provides a robust mechanism to handle correlated item structures, leading to superior performance in certain datasets.

3. The paper extensively evaluates DE and DEBC across various synthetic and real-world datasets, demonstrating their practical effectiveness and robustness across different settings.

**Weaknesses:**

1. While DE and DEBC perform well in practice, the theoretical sample complexity bounds provided in the paper are not as tight as their practical performance would suggest, leaving room for further theoretical refinement.

2. DEBC’s performance heavily relies on the strength of item correlations, this raise potential limiting its applicability in scenarios with weak correlations.

3. The paper primarily focuses on cosine similarity for item embeddings and correlations. Extending this to other similarity measures or more general settings is only briefly mentioned and not fully explored.

Some cosmetics: for example on row 330: `we can can combine`.

**Questions:**

How can the sample complexity bounds be further improved to match the practical performance observed in experiments, and is there potential for achieving instance-optimal sample complexity in the PAC best-item setting?

Could the proposed DE and DEBC algorithms be adapted or extended to work with partial or full rankings instead of just identifying the best item, and what challenges might arise in such extensions?

How would the algorithms perform in settings where item correlations are dynamic or evolve over time, and what adjustments to DE/DEBC might be necessary to handle such changes effectively?

---

### Meta-Review · Area_Chair_m9EE · 2024-12-16

**Metareview:**

This paper studies best item identification from top-$1$ feedback on a subset of items. The authors propose two algorithms for the problem, analyze them, and evaluate them empirically. They also study winning rate estimation of two items without directly comparing them. The scores of this paper are 6, 5, and 2x 3; and did not change during the discussion. The reviewers had several concerns:

* **Algorithmic novelty:** The idea of eliminating suboptimal items in learning to rank to find the best item / ranking is not new. In addition to the references pointed out by Reviewer DQ7C, I suggest that the authors look at [Online Learning to Rank in Stochastic Click Models](https://proceedings.mlr.press/v70/zoghi17a.html) and [TopRank: A Practical Algorithm for Online Stochastic Ranking](https://proceedings.neurips.cc/paper/2018/hash/de03beffeed9da5f3639a621bcab5dd4-Abstract.html).

* **Technical novelty:** The proof techniques are standard.

* **Theoretical improvement over SOTA:** The sample complexity bound is worse than in prior works, which indicates looseness.

These concerns require a major revision and therefore the paper cannot be accepted at this time.

**Additional Comments On Reviewer Discussion:**

See the meta-review for details.

---

### Decision · Program_Chairs · 2025-01-22

Reject